# Genome-wide identification of genes required for alternative peptidoglycan cross-linking in *Escherichia coli* revealed unexpected impacts of β-lactams

Henri Voedts[1], Sean P. Kennedy [2], Guennadi Sezonov[1], Michel Arthur [1,3] & Jean-Emmanuel Hugonnet [1,3]

The D,D-transpeptidase activity of penicillin-binding proteins (PBPs) is the well-known primary target of β-lactam antibiotics that block peptidoglycan polymerization. β-lactam-induced bacterial killing involves complex downstream responses whose causes and consequences are difficult to resolve. Here, we use the functional replacement of PBPs by a β-lactam-insensitive L,D-transpeptidase to identify genes essential to mitigate the effects of PBP inactivation by β-lactams in actively dividing bacteria. The functions of the 179 conditionally essential genes identified by this approach extend far beyond L,D-transpeptidase partners for peptidoglycan polymerization to include proteins involved in stress response and in the assembly of outer membrane polymers. The unsuspected effects of β-lactams include loss of the lipoprotein-mediated covalent bond that links the outer membrane to the peptidoglycan, destabilization of the cell envelope in spite of effective peptidoglycan cross-linking, and increased permeability of the outer membrane. The latter effect indicates that the mode of action of β-lactams involves self-promoted penetration through the outer membrane.

Gram-negative bacteria, such as *Escherichia coli*, possess a multilayer envelope that sustains the turgor pressure of the cytoplasm (cell wall peptidoglycan) and acts as a selective chemical barrier for nutrients, waste, and toxic compounds (inner and outer membranes) (Fig. 1a)[1]. Several polymers containing protein, peptide, glycan, and lipid moieties ensure the mechanical properties and the transport functions of the envelope. The peptidoglycan (PG), which is covalently linked to the outer membrane via the Braun lipoprotein, is a giant net-like macromolecule (*ca*. $10^9$ Da) polymerized from a disaccharide-pentapeptide unit (Fig. 1b). Glycosyltransferases catalyze the formation of β−1,4 glycosidic bonds between disaccharides to form glycan chains that are subsequently cross-linked to each other by transpeptidases (Fig. 1c).

The latter enzymes, the D,D-transpeptidases, are also referred to as penicillin-binding proteins (PBPs) as they are the essential targets of β-lactam antibiotics. For peptidoglycan polymerization, PBPs interact with the D-Ala⁴-D-Ala⁵ extremity of a pentapeptide stem (hence the D,D designation) and form a covalent link between D-Ala⁴ and their active-site Ser residue with the concomitant release of D-Ala⁵. In the following step, the resulting acyl-enzyme reacts with the second substrate, most often a tetrapeptide stem, to form a 4 → 3 cross-linked Tetra-Tetra dimer with the concomitant release of the PBP (Fig. 1c; Supplementary Fig. S1a)[2]. In cooperation with scaffolding proteins and endopeptidases, the D,D-transpeptidases mediate the insertion of glycan strands into the expanding PG net-like macromolecule, thereby determining

[1]Centre de Recherche des Cordeliers, Sorbonne Université, Inserm, Université Paris Cité, F-75006 Paris, France. [2]Institut Pasteur, Université Paris Cité, Département Biologie Computationnelle, F-75015 Paris, France. [3]These authors contributed equally: Michel Arthur, Jean-Emmanuel Hugonnet. e-mail: michel.arthur@crc.jussieu.fr; jean-emmanuel.hugonnet@crc.jussieu.fr

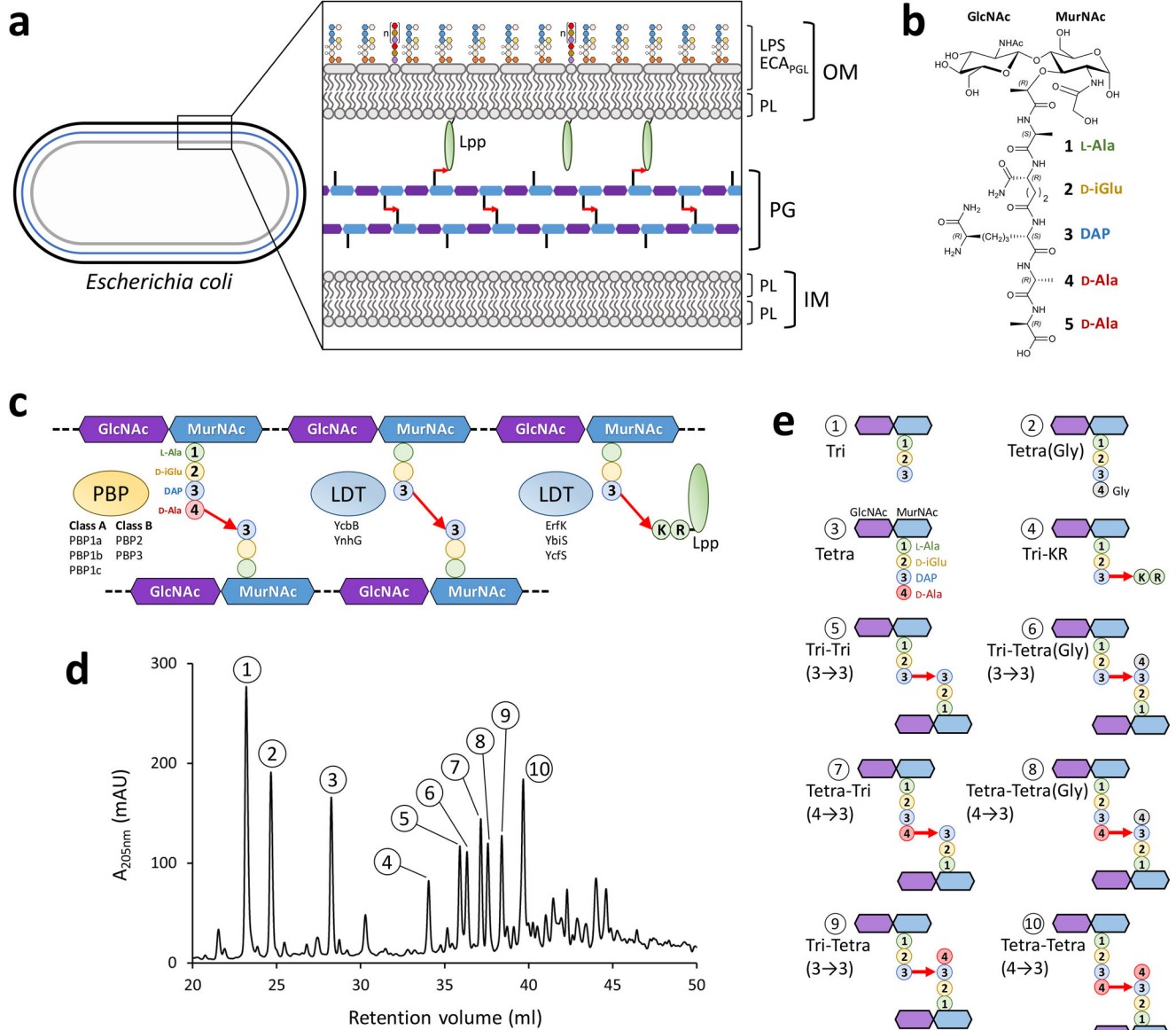

**Fig. 1 | Structure of the multilayered *E. coli* envelope. a** The envelope is composed of inner (IM, gray) and outer (OM, black) membranes. Peptidoglycan (PG, blue) ensures osmotic protection of the bacterial cell. The IM is a lipid bilayer composed mainly of phospholipids (PL). The OM is an asymmetric lipid bilayer: the inner leaflet is composed of PL and the outer leaflet of lipopolysaccharides (LPS) and phosphatidylglycerides substituted by the enterobacterial common antigen (ECA_{PGL}). **b** Structure of the PG subunit (GlcNAc, *N*-acetyl-glucosamine; MurNAc, *N*-acetyl-muramic acid). **c** PG is a net-like macromolecule made of glycan chains (blue-purple polygons) cross-linked together by short peptide stems (colored circles). PBPs catalyze the formation of 4 → 3 cross-links connecting the

4th position of an acyl donor stem to the 3rd position of the acceptor. Two members of the LDT family (YcbB and YnhG) catalyze the formation of 3 → 3 cross-links. Three LDTs (ErfK, YbiS, and YcfS) anchor the Braun lipoprotein (Lpp) to the PG. Linking DAP at the 3rd position of a PG donor stem to the Arg-Lys extremity of the Lpp provides a covalent link between the OM and the PG. The PG-Lpp link is hydrolyzed by the YafK LDT[46,47]. **d** *rp*HPLC profile of muropeptides from strain BW25113(*ycbB*, *relA'*) grown without ceftriaxone. Structure is inferred from the fragments obtained by digestion of PG with muramidases which cleave the MurNAc-GlcNAc β−1,4 bond. **e** Structure of the muropeptides deduced from mass spectrometry analyses. Source data are provided as a Source Data file.

the bacterial shape and ensuring a mechanical barrier against the osmotic pressure of the cytoplasm during the entire cell cycle[3–7].

Bypass of the D,D-transpeptidase activity of PBPs by structurally unrelated L,D-transpeptidases (LDTs) results in resistance to most β-lactams[8,9]. The L,D-transpeptidases cleave the L,D *meso*DAP3-D-Ala4 peptide bond of tetrapeptide stems and form 3 → 3 cross-links (Fig. 1c; Supplementary Fig. S1b)[10]. Other members of the L,D-transpeptidase family are responsible for the anchoring of the Braun lipoprotein to the PG (Supplementary Fig. S1c)[11]. Activation of the mostly cryptic L,D-transpeptidase PG polymerization pathway was first discovered in β-lactam-resistant mutants of *Enterococcus faecium* selected in vitro[12,13].

L,D-transpeptidation is the main mode of PG cross-linking in pathogenic *Mycobacterium tuberculosis*[14] and can be targeted by third-line chemotherapies based on the β-lactam meropenem[15]. In *E. coli*, activation of the pathway requires overproduction of both the YcbB L,D-transpeptidase (LdtD) and of the guanosine penta- or tetra-phosphate (p)ppGpp alarmone[8]. YcbB was subsequently reported to participate in PG maintenance as it compensated defects in lipopolysaccharide export[16].

The D,D-transpeptidase activity of PBPs was identified as the primary target of β-lactams in the 1960s[17]. Nevertheless, deciphering the downstream cascade of events that leads to bacterial death poses

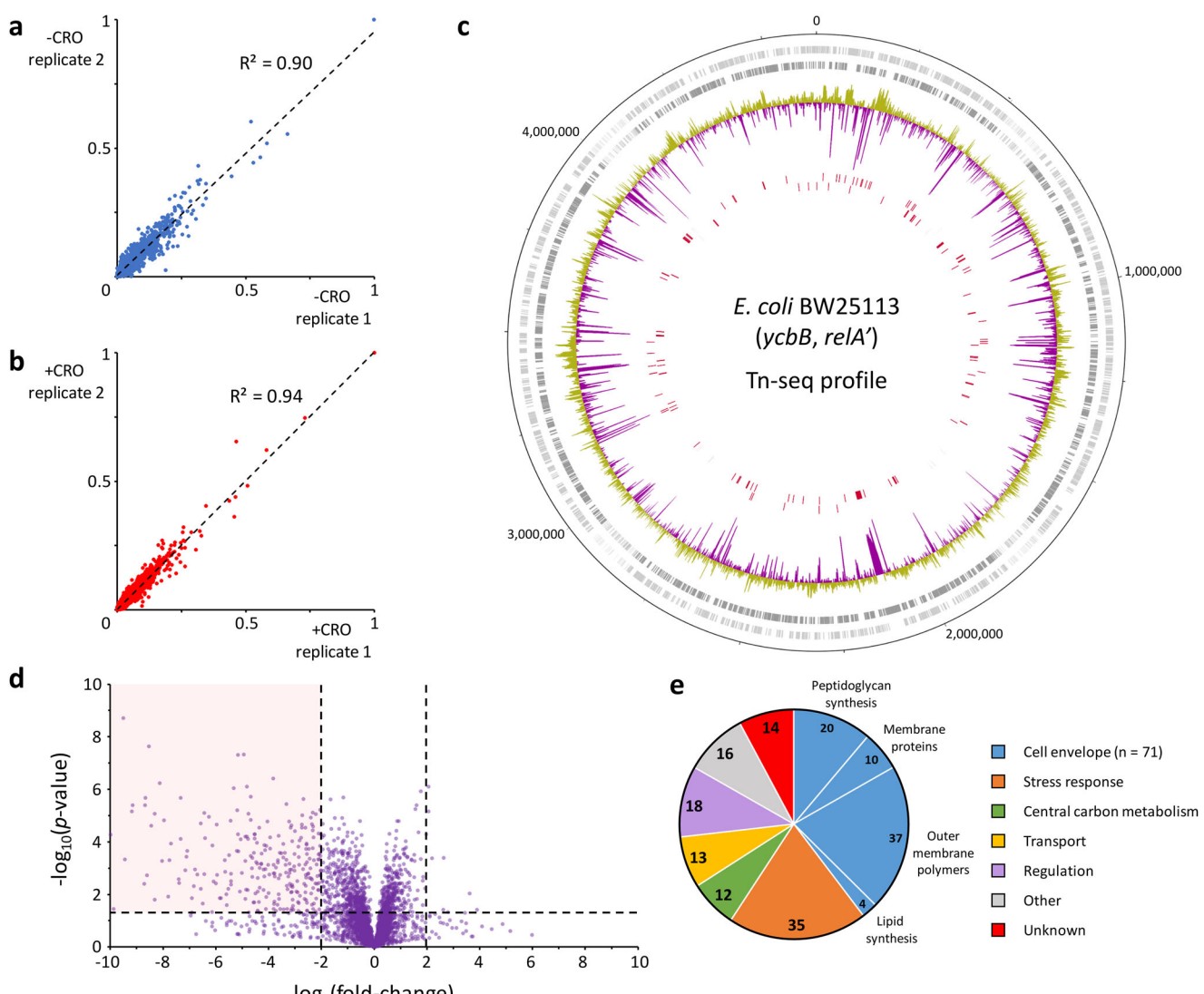

**Fig. 2 | Genome-wide transposon sequencing in *E. coli* BW25113(*ycbB, relA'*) grown in the absence (−CRO) or presence (+CRO) of ceftriaxone.**
**a**, **b** Correlation of normalized average read numbers per CDS for two sequenced technical replicates obtained for the −CRO and +CRO conditions, respectively. **c** Frequency and location of transposon insertions. Rings, outermost to innermost, correspond to: (i) the BW25113 coordinates; (ii) the sense (light gray) and antisense (gray) CDSs; (iii) the log₂ of the fold-change of insertion for the −CRO versus +CRO conditions. Data for −CRO to +CRO ratios smaller and greater than 1 appear in yellow and purple, respectively; and (iv) the sense and antisense CDSs specifically essential in the +CRO condition, respectively. **d** The volcano plot represents genes for which transposon inactivation is beneficial or detrimental in the +CRO condition. The *X*-axis represents the log₂ of the fold-change of the average reads per gene for the −CRO *versus* +CRO conditions. The *Y*-axis represents the negative log₁₀ of the *p*-value obtained from unpaired two-tailed *t*-test. The pink area corresponds to genes with an at least 4-fold lower number of insertions in the +CRO condition with a *p*-value < 0.05. **e** Biological functions of genes specifically essential in the +CRO condition. Source data are provided as a Source Data file.

particular challenges[18–21]. This topic remains the subject of intense investigations.

In this work, we investigate the functions that are essential for the rescue of the β-lactam-inactivated 4 → 3 cross-linking activity of PBPs by the 3 → 3 cross-linking activity of ʟ,ᴅ-transpeptidase YcbB at the genome scale. Results are based on Tn-seq identification of genes essential for YcbB-mediated β-lactam resistance, construction of multiple gene deletions to decipher epistatic relationships, and determination of the structure of PG by mass spectrometry (Fig. 1d, e). This analysis opens a new window on the evolvability of the cell envelope, on the complex network of interactions that pertain to the homeostasis of envelope polymers, and on the mechanisms that *E. coli* develops to fight against the stress induced by β-lactam-mediated ᴅ,ᴅ-transpeptidase inactivation.

## Results

### Identification of genes essential for YcbB-mediated β-lactam resistance

A transposon sequencing (Tn-seq)[22] approach was used to identify genes essential for YcbB-mediated β-lactam resistance in the genome of *E. coli* BW25113(*ycbB, relA'*)[8]. This strain combines high level production of ʟ,ᴅ-transpeptidase YcbB and of (p)ppGpp synthetase RelA' upon induction by IPTG and ʟ-arabinose of the *ycbB* and *relA'* genes, respectively. Two experimental conditions, absence (−CRO) or presence (+CRO) of ceftriaxone, were compared. The β-lactam ceftriaxone was chosen for our analysis since this broad spectrum 3rd generation cephalosporin is highly effective in inactivating all ᴅ,ᴅ-transpeptidases belonging to the PBP family and is specific of these enzymes as ʟ,ᴅ-transpeptidases are not inactivated due to a combination of ineffective acylation and formation of an acyl-enzyme prone to

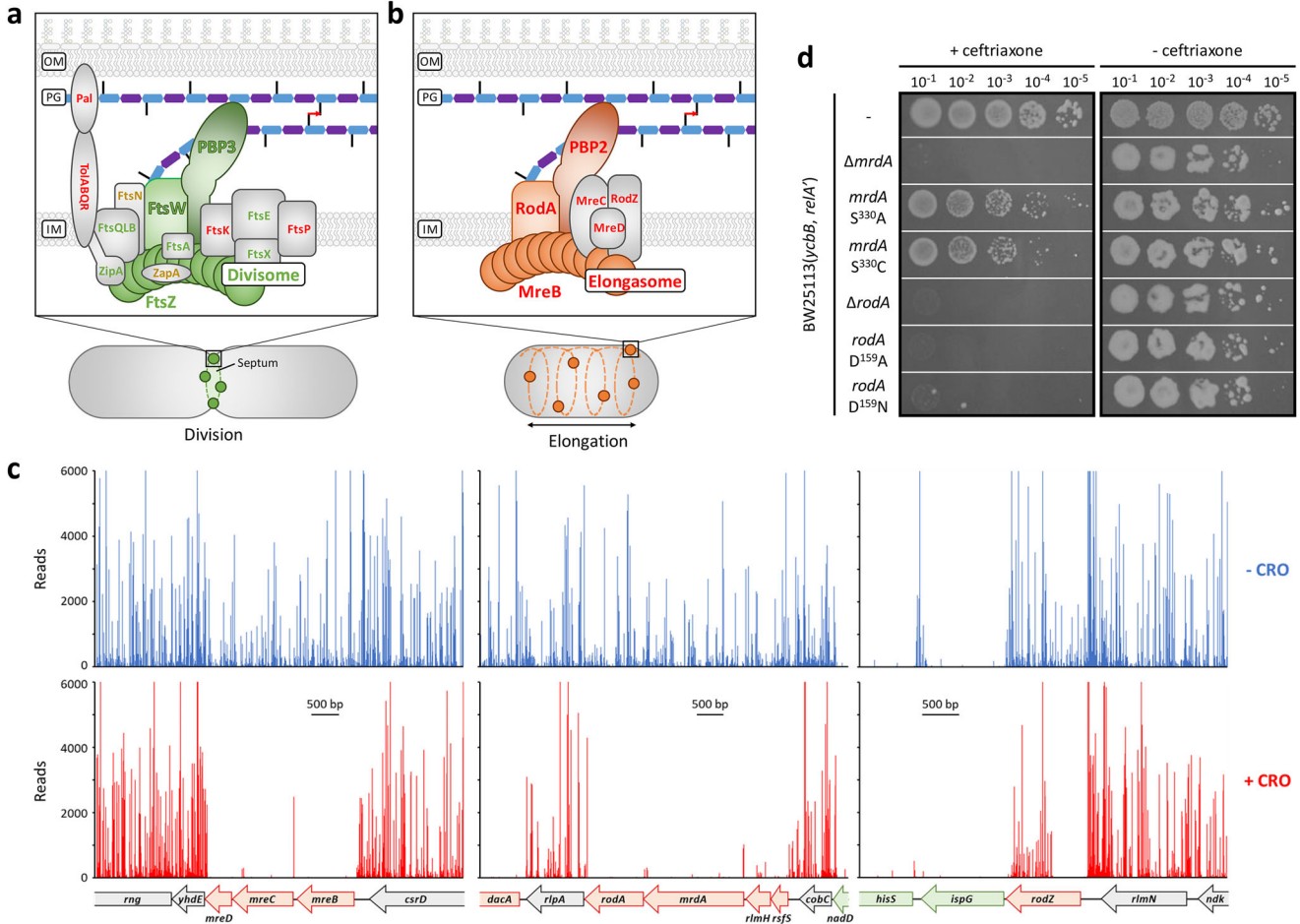

**Fig. 3 | Tn-seq evaluation of genes encoding components of the PG polymerization complexes. a**, **b** Schematic representation of the divisome and elongasome, respectively. **c** Tn-seq insertion profiles of genes encoding components of the elongasome. Reading frames are indicated at the bottom of the panel and color coded: red, genes selectively essential for +CRO; green, genes essential for both −CRO and +CRO; gray, genes non-essential in either condition. Transposon insertion sites are indicated by lines above the reading frames with their height reflecting the number of reads for each insertion. **d** Essentiality of genes encoding elongasome components PBP2 and RodA tested by site-directed mutagenesis. Plating efficiency was tested in the presence of ceftriaxone at 8 μg/ml (+ceftriaxone) or without drug (−ceftriaxone) on BHI agar plates supplemented with 40 μM IPTG and 1% ʟ-arabinose for induction of *ycbB* and *relA'*, respectively. Plates were incubated at 28 °C since the strains harboring the *mrdA* S$^{330}$A and *mrdA* S$^{330}$C did not grow in the presence of ceftriaxone at 37 °C.

hydrolysis[23,24]. For both conditions, IPTG and ʟ-arabinose were added in order to induce genes encoding the YcbB ʟ,ᴅ-transpeptidase and the RelA' (p)ppGpp synthase, respectively. Genomic DNA extraction was performed on pools of 810,000 (−CRO) and 260,000 (+CRO) transformants and sequenced. Technical duplicates of independent DNA extractions revealed $R^2$ correlation coefficients of 0.90 and 0.94 for the −CRO and +CRO conditions, respectively (Fig. 2a, b). Junction-containing sequences were aligned to the *E. coli* BW25113 reference genome and essential genes identified by quantifying the presence of gaps in the insertions (Fig. 2c). The average numbers of reads per gene were compared to identify genes in which transposon insertions were significantly (*p*-value < 0.05) underrepresented in the +CRO condition (Fig. 2d; Supplementary Data File 1a).

The Tn-seq analyses revealed that among the 4309 annotated CDSs, 179 were selectively essential in the +CRO condition according to a stringent combination of three criteria including (i) the presence of a gap in transposon insertions, (ii) a significant *p*-value (< 0.05), and (iii) a fold-change larger than 4 (see "Methods") (Supplementary Data File 1b). No genes were found to be selectively essential in the −CRO condition. The 179 +CRO selectively essential genes were assigned to seven functional categories with a majority found to be involved in the metabolism of envelope components (*N* = 71, 40%) and in stress responses (*N* = 35, 20%) (Fig. 2e). Relevant Tn-seq profiles of genes

discussed in this study have been compiled in Supplementary Data File 2.

## Divisome disturbances are not tolerated for YcbB-mediated β-lactam resistance

Synthesis of the septum is mediated by components of the divisome that are recruited at mid-cell by the assembly of the FtsZ ring (Fig. 3a)[6]. Proteins of the core divisome, FtsA, FtsEX, FtsQLB, FtsW, FtsZ, ZipA, and PBP3, were each essential both in the −CRO and +CRO conditions. In contrast, divisome-associated proteins involved in the control of the cell cycle (MinCD, FtsK)[25,26], in the stabilization of the divisome under stress conditions (FtsP)[27–29], and in providing a physical link with the outer membrane (TolABQR and Pal)[30] were only essential in the +CRO condition (Supplementary Data File 1b; Supplementary Data File 2).

## Glycan chain polymerization by the elongasome is required for YcbB-mediated β-lactam resistance

Expansion of the side-wall of peptidoglycan is mediated by the elongasome (Fig. 3b), which includes (i) RodA, a glycosyltransferase belonging to the shape, elongation, division, and sporulation (SEDS) protein family, (ii) PBP2, a catalytically mono-functional class B PBP with ᴅ,ᴅ-transpeptidase activity, and (iii) MreB, an actin-like protein forming a scaffolding cytoskeleton for the elongasome complex[6].

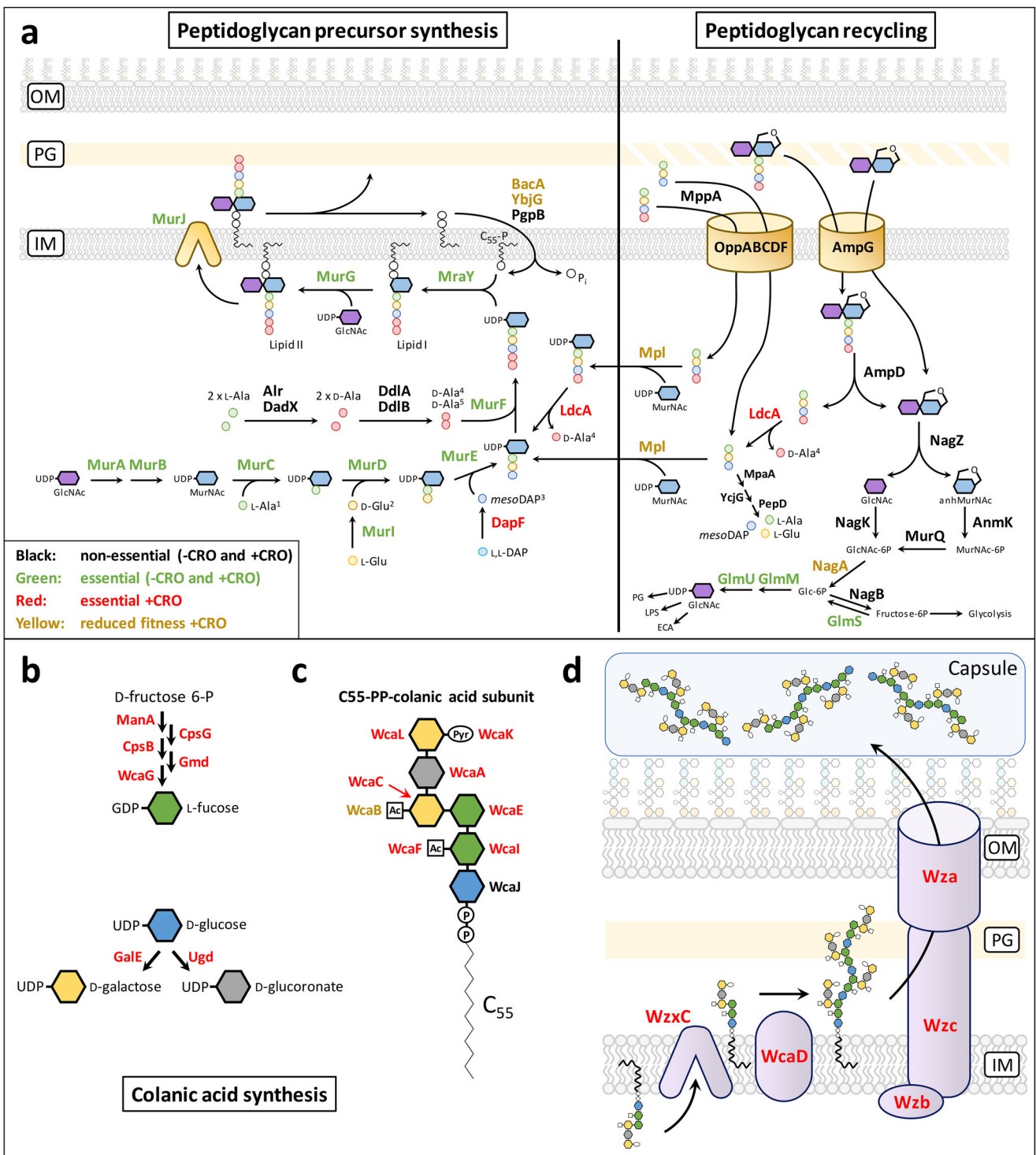

**Fig. 4 | Essentiality of genes involved in PG recycling and colanic acid bio-synthesis. a** The PG subunit is assembled both by de novo synthesis (left) and recycling of PG fragments (right). **b** Synthesis of colanic acid nucleotide sugar precursors. **c** Assembly of the colanic acid subunit linked to the undecaprenyl lipid carrier. **d** Polymerization and transport of colanic acid to the cell surface. Abbreviations: Ac acetyl; $C_{55}$ undecaprenyl lipid carrier; IM inner membrane; OM outer membrane; P phosphate; Pyr pyruvate.

Previous analyses established that the elongasome is dispensable for growth in mutants overproducing the (p)ppGpp alarmone[31,32]. As expected, Tn-seq analysis indicated that each of the six elongasome genes (*mrdA*, *rodA*, *rodZ*, *mreB*, *mreC*, and *mreD*) were dispensable for growth in the −CRO condition, which included L-arabinose for production of (p)ppGpp by RelA' (Fig. 3c). Growth of the elongasome mutants was further dependent upon expression of the *relA'* (p)ppGpp synthase gene (Supplementary Fig. S2a). Strikingly, the six elongasome genes were each essential in the +CRO condition despite (p)ppGpp production (Fig. 3c). The essentiality of PBP2 (encoded by *mrdA*) was unexpected given that this PBP could not participate directly in PG cross-linking owing to inactivation of its D,D-transpeptidase domain by ceftriaxone[8,33]. The components of the elongasome are encoded by compact gene clusters that are highly sensitive to polar effects

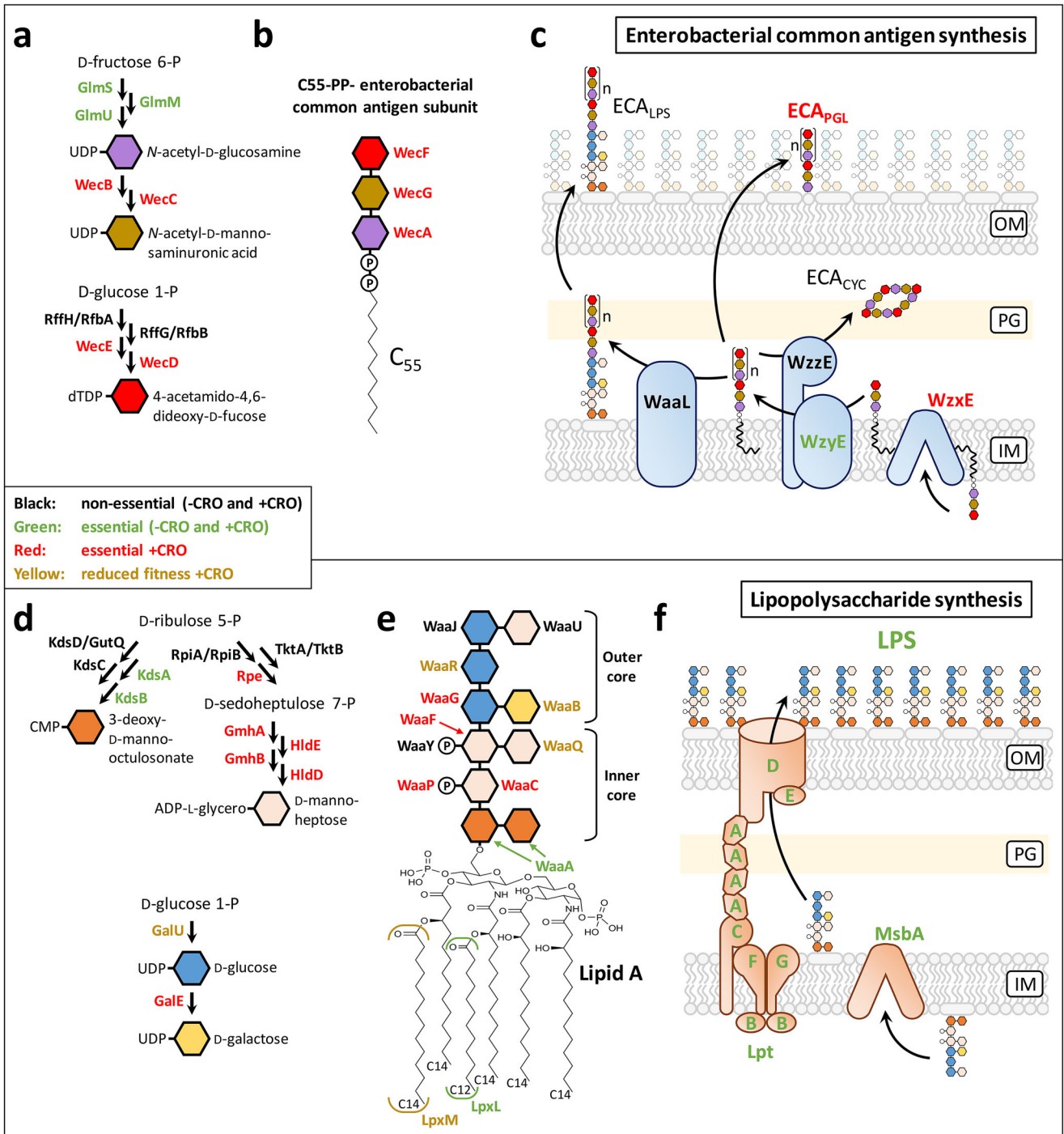

**Fig. 5 | Enterobacterial common antigen (ECA) and lipopolysaccharide (LPS) biosynthesis. a** Synthesis of ECA nucleotide-sugar precursors. **b** Assembly of the ECA subunit linked to the undecaprenyl lipid carrier. **c** Polymerization and transport of ECAs to the cell surface. **d** Synthesis of core LPS nucleotide-sugar precursors. **e** Assembly of the core LPS. **f** LPS transport to the cell surface. Abbreviations: $C_{55}$ undecaprenyl lipid carrier; IM inner membrane; OM outer membrane; P phosphate.

potentially altering the transcription levels of adjacent genes. We therefore complemented our Tn-seq results using a CRISPR-Cas9 approach (Supplementary Fig. S2b) for one-step site-directed mutagenesis of *mrdA* to impair the transpeptidase activity of PBP2. Expression of resistance to ceftriaxone was abolished by deletion of *mrdA*, but not by replacement of the catalytic Ser residue of the D,D-transpeptidase by Ala or Cys (Fig. 3d; Supplementary Fig. S2c). Thus PBP2, but not its transpeptidase activity, was essential for drug resistance.

A similar CRISPR-Cas9 approach, applied to the mutagenesis of the glycosyltransferase gene *rodA*, revealed that a catalytically active form of the glycosyltransferase was required for growth in the presence of ceftriaxone (Fig. 3d; Supplementary Fig. S2c). It was previously shown that PBP2, RodZ, MreB, MreC, and MreD are each essential for the assembly of the elongasome, and that the interaction between PBP2 and RodA is not dependent on PBP2 activity[34–36]. Thus, PG cross-linking by YcbB required polymerization of glycan chains by RodA in a functional elongasome containing both RodA

and PBP2, with the transpeptidase activity of the latter enzyme being dispensable.

## Recycling of peptidoglycan fragments is dispensable for YcbB-mediated β-lactam resistance

In *E. coli*, 40–50% of the PG is degraded per generation with > 90% of the resulting muropeptides recycled by the cell[37,38]. This involves the retrograde transport of PG fragments into the cytoplasm by specialized permeases, AmpG and the Opp complex, followed by the recycling of the sugar and peptide moieties as glucosamine and tripeptide L-Ala-γ-D-Glu-DAP, respectively (Fig. 4a). Tn-seq analysis showed that genes encoding the AmpG permease and each component of the Opp complex were fully dispensable for growth both in −CRO and +CRO conditions. We further demonstrated that PG recycling was fully dispensable by constructing a Δ*ampG* Δ*oppF* Δ*mppA* triple mutant that was found to be viable and resistant to ceftriaxone in spite of the loss of both transporters (Supplementary Fig. S3a). Nevertheless, the cytoplasmic L,D-carboxypeptidase LdcA was essential in the +CRO but not in the −CRO condition (Supplementary Data File 1b; Supplementary Data File 2). This enzyme was previously shown to trim D-Ala[4] from recycled tetrapeptide stems to provide the L-Ala-γ-D-Glu-DAP tripeptide that is directly added to UDP-MurNAc by the Mpl ligase[39,40]. Disruption of the gene encoding LdcA was reported to result in bacteriolysis during stationary growth, attributed to dramatically decreased cross-linking as recycled tetrapeptide stems cannot serve as acyl donors by D,D-transpeptidases (Supplementary Fig. S1)[40]. The same explanation cannot account for the essential role of LdcA in ceftriaxone resistance since YcbB uses tetrapeptide stems as acyl donors[8]. Nonetheless, the essential role of LdcA in the +CRO condition stemmed from elimination of imported tetrapeptides since deletion of the *mpl* gene restored growth of the Δ*ldcA* mutant in the presence of ceftriaxone (Supplementary Fig. S3a). Overexpressing *murA*, encoding the enzyme catalyzing the first committed step of PG synthesis (Fig. 4a), also restored resistance of the Δ*ldcA* mutant (Supplementary Fig. S3b). Thus, boosting the metabolic flux trough the PG assembly pathway compensates for the toxicity of the imported tetrapeptides. Together, these results indicate that Mpl ligates the tetrapeptide onto UDP-MurNAc and that in the absence of LdcA the resulting tetrapeptide-containing precursors are ineffectively used in subsequent PG synthesis steps and impair the global efficacy of the pathway.

## Consequences of replacement of *meso*DAP by L,L-DAP in PG precursors on resistance and envelope synthesis

DapF, which converts L,L-DAP into *meso*DAP, was reported to be dispensable for growth in *E. coli*[41]. MurE ligase catalyzes the addition of L,L-DAP to UDP-MurNAc-L-Ala-D-Glu, albeit less effectively than the addition of *meso*DAP due to a 3000-fold difference in $K_m$[41]. Deletion of the gene encoding DapF results in the incorporation of L,L-DAP into peptidoglycan precursors. L,L-DAP-containing stem peptides were ineffectively used as acyl donors by the PBPs but not as acceptor[41]. The overall cross-linking of peptidoglycan was moderately reduced[41]. Concordantly, L,L-DAP was strongly discriminated against *meso*DAP by purified PBP1b in vitro[42]. In our study, *dapF* was non-essential in the −CRO condition. On the contrary, no transposon insertions were found in this gene in the +CRO condition (Supplementary Data File 1b; Supplementary Data File 2). Formation of 3→3 cross-linked dimers was detected in a Δ*dapF* mutant (Supplementary Fig. S4) but mass spectrometry analysis did not enable discriminating between the two DAP stereoisomers. Our peptidoglycan analysis additionally showed that deletion of *dapF* did not prevent the anchoring of the Braun lipoprotein to the PG (Supplementary Fig. S4). Thus, L,D-transpeptidases, similarly to PBPs, tolerated to a certain extent the incorporation of L,L-DAP into peptidoglycan precursors following deletion of *dapF*. The essentiality of *dapF* in the +CRO condition might be accounted for by a

reduced efficacy of the PG assembly pathway, as observed above for the *ldcA* null mutant.

## Essentiality of redundant and non-redundant peptidoglycan biosynthetic enzymes

Excepting *dapF*, all genes encoding enzymes involved in PG biosynthetic steps known to be performed by a single enzyme were identified as essential in both +CRO and −CRO conditions (Fig. 4a). Data for essential biosynthetic steps involving redundant enzyme functions are reported in Supplementary Fig. S5.

## Accumulation of capsule intermediates abolished YcbB-mediated β-lactam resistance

The outermost polymer of the cell wall of most *E. coli* strains is a capsule made of colanic acid[43]. WcaJ catalyzes the first committed step in the assembly of the colanic acid precursor. The gene encoding WcaJ was non-essential in the −CRO and +CRO conditions, indicating that the absence of the capsule was compatible for growth both in the absence or presence of ceftriaxone (Fig. 4b–d). Unexpectedly, all other enzymes in the colanic acid synthesis pathway were essential in the +CRO but not in the −CRO conditions (Supplementary Data File 1b; Supplementary Data File 2). Given that the loss of all other colanic acid biosynthesis enzymes is expected to result in accumulation of undecaprenyl-linked precursors, we concluded that these enzymes are essential for preventing the sequestration of the lipid carrier. According to this model, sequestration of $C_{55}$ would indirectly inhibit the assembly of other polymers, such as the essential PG, that rely on the same lipid carrier. As confirmation, we showed that deletion of the gene encoding WcaJ, catalyzing the first committed step of the colanic acid precursor synthesis, restored the growth of a WcaI mutant in the presence of ceftriaxone (Supplementary Fig. S6a). Combined, these results indicate that the capsule is, in itself, non-essential for growth in the presence of ceftriaxone although colanic acid biosynthetic enzymes are essential to prevent reduced PG synthesis by sequestration of the lipid carrier.

## Outer-membrane enterobacterial common antigens (ECAs) are required for YcbB-mediated β-lactam resistance

Three ECA polymers are assembled from the same $C_{55}$-linked precursor (Fig. 5)[44]. ECAs composed of 1 to 14 units of the precursor are covalently linked to lipopolysaccharides ($ECA_{LPS}$) or to phosphatidylglycerides ($ECA_{PGL}$) in the outer leaflet of the outer membrane. Free cyclic ECAs ($ECA_{CYC}$), made of four units, are located in the periplasm.

None of the three ECA polymers were essential in the −CRO condition since transposon insertions were detected in genes encoding enzymes involved in the synthesis of nucleotide-sugar precursors (Fig. 5a) and for the assembly of the $C_{55}$-linked trisaccharide subunit (Fig. 5b). In contrast, most enzymes were essential in the +CRO condition indicating that at least one of the three ECA polymers was essential for resistance (Supplementary Data File 1b; Supplementary Data File 2). Synthesis of both $ECA_{LPS}$ and $ECA_{CYC}$ was dispensable for resistance since deletion of the genes encoding the WaaL and WzzE ligases, alone or in combination, was dispensable for growth in the presence of ceftriaxone (Fig. 5c; Supplementary Fig. S6b). The phenotype of the double mutant indicates the absence of redundancy between the $ECA_{LPS}$ and $ECA_{CYC}$ for ceftriaxone resistance. By elimination, these results indicate that synthesis of $ECA_{PGL}$ is essential for resistance although this could not be directly established as the corresponding polymerase has not yet been identified.

## Synthesis of the core sugars of lipopolysaccharides (LPS) is essential for YcbB-mediated β-lactam resistance

Lipid A, substituted by two 3-deoxy-D-manno-octulosonate residues, is known to be the minimal moiety of LPS required for growth of *E. coli*[45]. Accordingly, most enzymes involved in its synthesis and

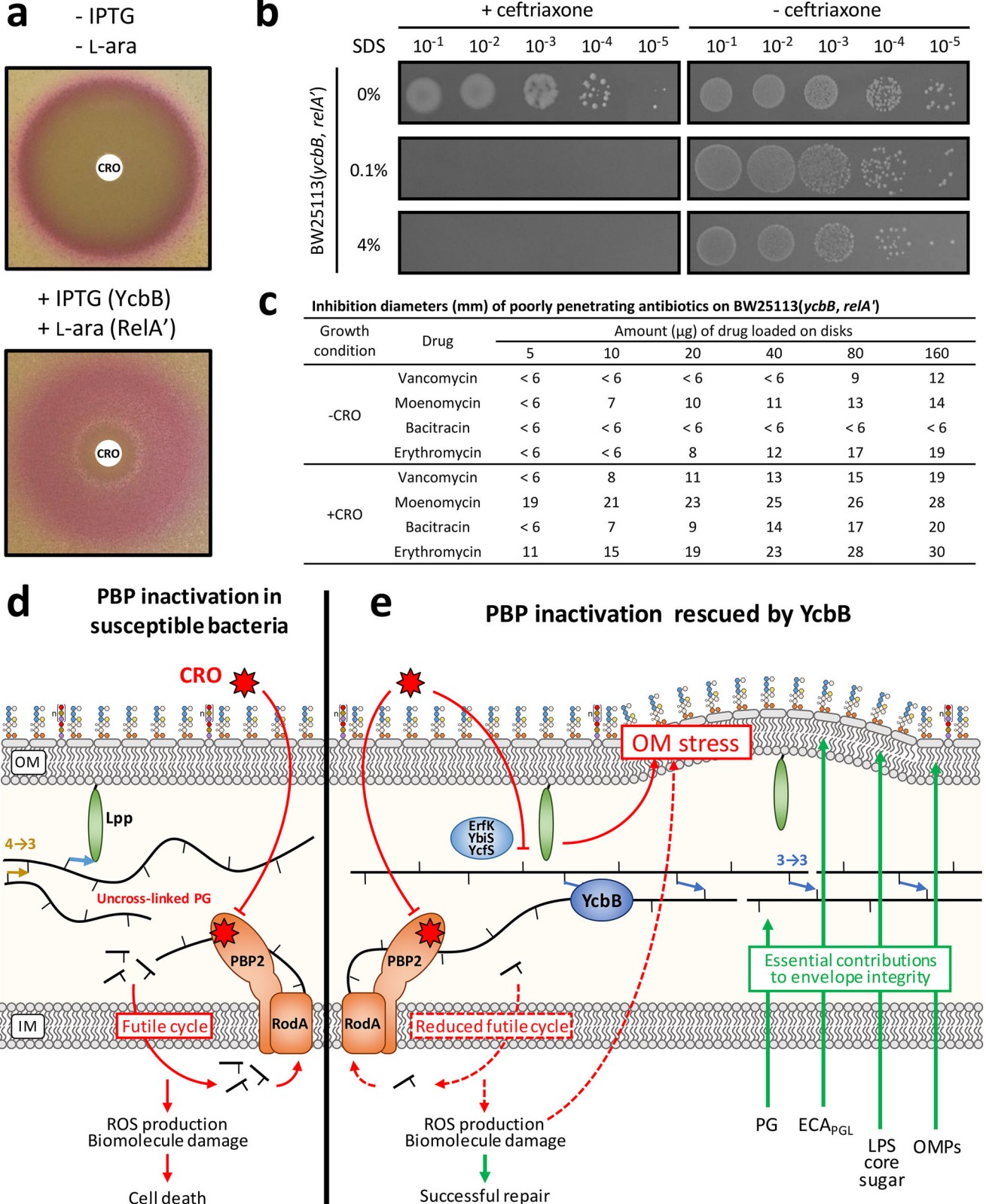

**Fig. 6 | Testing outer membrane permeability in the presence of ceftriaxone.**
**a** Hydrolysis of chromogenic CPRG at 20 μg/ml by cytoplasmic LacZ[63]. The disks were loaded with 30 μg ceftriaxone. **b** Plating efficiency in the presence of SDS. Growth was tested in the presence of ceftriaxone at 8 μg/ml (+ceftriaxone) or in the absence of the drug (−ceftriaxone) on BHI agar plates supplemented with 40 μM IPTG and 1% L-arabinose for induction of *ycbB* and *relA'*, respectively.
**c** Inhibition diameters of drugs that do not penetrate the outer membrane of wild-type *E. coli*. Values are the median of three biological repeats. **d** Cascade of events leading to bacterial cell death due to inactivation of PBPs by β-lactams. **e** Rescue of PBP inactivation by the YcbB L,D-transpeptidase. Abbreviations: IM inner membrane; OM outer membrane; Lpp Braun lipoprotein; OMPs outer membrane proteins. Source data are provided as a Source Data file.

transport to the outer membrane were essential in the −CRO and +CRO conditions (Fig. 5d–f). Downstream enzymes, responsible for the synthesis of the ADP-L-glycero-D-manno-heptose subunit (RpiA, Rpe, GmhA, HldE, GmhB, and HldD), the addition of a D-manno-heptose to the saccharide chain (WaaC), and the phosphorylation of this residue (WaaP), were essential only in the +CRO condition. Additions of a 2nd D-manno-heptose (WaaF) and a D-glucose (WaaG) residues were also essential for resistance (Supplementary Data File 1b; Supplementary Data File 2). The enzymes catalyzing the three subsequent steps in the assembly of the outer core (WaaQ, B, and R) were required for optimal growth in the +CRO condition. Only three enzymes of the pathway (WaaY, J, and U) were fully dispensable both in the −CRO and +CRO conditions. These results indicate that growth in the presence of ceftriaxone was not compatible with incomplete synthesis of LPS.

### The essentiality of *ldcA*, *dapF*, ECA_{PGL}, and LPS core sugars did not depend upon overproduction of capsular colanic acids

The spot assays revealed that growth in the +CRO condition was associated with a mucoid phenotype, a phenotype known to result from overproduction of colanic acids by stressed cells. This observation raises the formal possibility that the essentiality of *ldcA*, *dapF*, ECA_{PGL}, and LPS core sugars in the +CRO condition (above) could depend upon overproduction of colanic acids by bacteria exposed to ceftriaxone. This was not the case since *ldcA*, *dapF*, *wecA*, and *waaC* remained essential in Δ*wcaJ* strains deficient in colanic acid synthesis (Supplementary Fig. S6c).

### Growth with β-lactams impacts the anchoring of the Braun lipoprotein to peptidoglycan

The *E. coli* genome encodes six members of the LDT protein family (Supplementary Fig. S1). YcbB and YnhG catalyze the formation of 3→3 cross-links whereas ErfK, YbiS, and YcfS are responsible for covalently linking the Braun lipoprotein (Lpp) to PG[10,11]. The remaining enzyme, YafK, hydrolyzes the Lpp-PG link formed by ErfK, YbiS, and YcfS, thereby releasing the lipoprotein[46,47]. The Tn-seq analysis showed that the Braun lipoprotein is dispensable for growth in both −CRO and +CRO conditions. Analyses of the PG structure revealed that growth in the presence of ceftriaxone dramatically reduced the anchoring of the Braun lipoprotein to PG (Supplementary Fig. S7). Western blot analyses showed that Lpp synthesis was not reduced in the +CRO condition (Supplementary Fig. S7g). Thus, ceftriaxone may inhibit the anchoring reaction catalyzed by LDTs (Supplementary Fig. S1), but this inhibition is unlikely to be direct since LDTs are not effectively inhibited by β-lactams, except those belonging to the carbapenem class[23,24].

Any hypothesis on the mode of inhibition of Lpp anchoring to PG by β-lactams should take into consideration the fact that LDTs for PG cross-linking (YcbB and YnhG) and for Lpp anchoring (ErfK, YbiS, and YcfS) interact with the same donor substrate, a tetrapeptide donor stem, in the first catalytic step common to the two types of L,D-transpeptidation reactions (Supplementary Fig. S1b, S1c)[12]. In the second step, the resulting acyl enzymes react either with a peptidoglycan stem to form 3→3 cross-links or with Lpp to covalently link the lipoprotein to PG. The common substrate of the first step of the transpeptidation reactions implies that competition of LDTs for the same tetrapeptide donor, as previously proposed[47], may contribute to limiting the anchoring of the Braun lipoprotein to PG. However, this appears unlikely in the context of YcbB-mediated ceftriaxone resistance since the high proportion of 3→3 cross-links in the +CRO condition (40%) was not detrimental to the anchoring of Lpp (Supplementary Fig. S7). Thus, impaired Lpp anchoring in the presence of ceftriaxone may result from another indirect inhibitory effect of ceftriaxone on the activity of ErfK, YbiS, and YcfS, such as the formation of their

tetrapeptide substrate by a partner enzyme, or the stimulation of the release of the lipoprotein by YafK.

### Growth with β-lactams impacts the outer membrane permeability barrier

Having assessed the role of outer membrane components on the integrity of the outer membrane in the presence of ceftriaxone, our next objective was to determine its functionality as a permeability barrier. In the absence of inducers of *ycbB* and *relA'*, hydrolysis of chlorophenol red-β-D-galactopyranoside (CPRG) by the cytoplasmic LacZ β-galactosidase was detected in a narrow ring at the perimeter of the ceftriaxone inhibition zone (Fig. 6a). Thus, exposure to sub-inhibitory concentrations close to the minimal inhibitory concentration (MIC) of the drug resulted in increased permeability as there is no specific transporter for CPRG. Upon induction of *ycbB* and *relA'*, LacZ activity was detected in a large zone showing that ceftriaxone increased permeability at concentrations that were much lower than the MIC of that strain. Growth in the presence of ceftriaxone sensitized the bacteria to sodium dodecyl sulfate (SDS) resulting in a > 40-fold reduction of the MIC of the detergent (Fig. 6b). Growth in the presence of ceftriaxone also increased susceptibility to vancomycin, moenomycin, bacitracin, and erythromycin, which are known to poorly penetrate the outer membrane of Gram-negative bacteria (Fig. 6c). Together, these data indicate that bypass of PBPs by YcbB was associated with the permeabilization of the outer membrane in the presence of ceftriaxone. This implies that the mode of action of β-lactams may involve a positive loop in which binding of the drugs to their targets promotes drug access to the periplasm. Damage to the outer membrane mediated by reactive oxygen species (ROS) may be involved in this process since increased lipid peroxidation was detected by a colorimetric assay (Supplementary Fig. S7h).

## Discussion

The targets of β-lactams and the mechanism of their inactivation have been known for decades, and yet an understanding of the downstream cascade of events that leads to bacterial cell death remains poorly-understood. The most recent model (Fig. 6d) proposes that the inactivation of PBPs results in the production of uncross-linked glycan chains in the periplasm and in their subsequent cleavage by lytic enzymes[19,21]. The anabolic demand generated by this futile cycle of PG synthesis and degradation engenders alterations in central carbon metabolism, protein synthesis, and ATP utilization leading to the production of ROS that damage biomolecules, including DNA, RNA, proteins, and lipids. The resulting oxidative stress contributes to cell death[18,48].

The rescue of PBP inactivation by YcbB implies, in its simplest sense, a bypass of the cross-linking activity of PBPs by the replacement of 4→3 by 3→3 cross-links (Fig. 1). However, our analysis shows β-lactam resistance to be significantly more complex since the toxic effects of the inactivation of PBPs persisted despite the bypass of their D,D-transpeptidase activity by YcbB. In agreement, increased lipid peroxidation was detected in the +CRO condition (Supplementary Fig. S7h). The genes identified as selectively essential in the +CRO condition (Fig. 2; Supplementary Data File 1b) revealed the complexity of functions that are mobilized to mitigate these toxic effects (Fig. 6e).

Uncoupling of RodA-mediated transglycosylation from PBP2-mediated transpeptidation upon inactivation of this PBP by β-lactams was previously shown to have opposite effects on the viability of susceptible and resistant bacteria. In susceptible bacteria, accumulation of uncross-linked glycan strands following PBP2 inactivation by β-lactams is deleterious and their degradation by the lytic transglycosylase SltY contributes to cell survival[21]. In contrast, the inactivation of SltY favors YcbB-mediated resistance to β-lactams, with uncross-linked glycan strands formed by the elongasome complex serving as suitable substrates for the assembly of a functional PG polymer[7]. Pulse-labeling

experiments indicated that YcbB supported PG polymerization via a two-step mechanism involving (i) assembly of 3 → 3 cross-linked glycan strands followed by (ii) insertion of the resulting neo-synthesized polymer into the existing PG[49]. In the present work, we provide direct evidence that this mode of peptidoglycan polymerization requires a functional elongasome, supporting glycan strand polymerization by catalytically active RodA (Fig. 3).

PG recycling is stimulated in the +CRO condition[49]. This implies that the rescue of PBPs by YcbB reduces but does not abolish the high anabolic demand necessary to feed the PG assembly pathway in the presence of the drug. Here we show that reducing the metabolic flux of precursors in the PG assembly pathway is not compatible with YcbB-mediated β-lactam resistance. This was observed to be the case for *dapF* or *ldcA* deletions, which led to the synthesis of aberrant precursors containing L,L-DAP or a tetrapeptide stem, respectively (Fig. 4a). This was also the case for inactivation of genes encoding colanic acid synthesis enzymes that led to the sequestration of the undecaprenyl lipid carrier competitively used for PG synthesis (Fig. 4b–d). The residual excess in the anabolic demand may also have consequences not directly linked to the efficacy of PG synthesis. Indeed, an elevated anabolic demand implies that ROS production remains elevated. The successful defense against the resulting oxidative stress may account for the selectively essential roles of regulators, chaperones, and enzymes involved in the repair of macromolecules (Fig. 2e; Supplementary Data File 1b).

Surprisingly, we found that resistance also depended on ECA_{PGL} (Fig. 5a), LPS bearing a full core sugar (Fig. 5b–d), on components of the Tol-Pal system (Fig. 3a), and on outer membrane proteins including OmpA and OmpC (Supplementary Data File 1b; Supplementary Data File 2). Bacterial morphology analyses recently showed that the outer membrane directly contributes to the mechanical properties of the cell envelope, a role historically attributed solely to the PG layer[50,51]. Strikingly, the stabilizing components essential for the stiffness of the outer membrane identified in one of these previous reports[50] included LPS bearing a full core sugar, the Tol-Pal system, and OmpA, also identified here as essential for YcbB-mediated ceftriaxone resistance. Thus, assembly of a functional envelope in the presence of ceftriaxone required both the bypass of the D,D-transpeptidase activity of PBPs by the L,D-transpeptidase activity of YcbB and the insertion of critical polymers in the outer membrane.

In conclusion, the rescue of PBPs by YcbB, coupled to a genome-wide Tn-seq screen, revealed two unexpected impacts of β-lactams. First, exposure to ceftriaxone prevented L,D-transpeptidase-mediated anchoring of the Braun lipoprotein to PG (Supplementary Fig. S7). Second, exposure to ceftriaxone increased the permeability of the outer membrane (Fig. 6). The latter effect implies that self-promoted penetration through the outer membrane is a previously unrecognized facet of the mode of action of β-lactams. In addition, our data show that the β-lactam-induced futile cycle was only sustainable if the efficacy of the peptidoglycan assembly pathway was not compromised by ineffective biosynthetic steps or by competition with the assembly of other polymers relying on the common undecaprenyl lipid carrier. The PG cross-linked by YcbB was functional although the integrity of the cell envelope heavily relied on outer membrane polymers that were otherwise dispensable for growth. Thus, identification of the genes involved in the rescue of PBPs by YcbB provided a positive screen to identify bacterial responses that prevent bacterial killing by β-lactams. In contrast, metabolic analyses restricted to susceptible bacteria are insensitive to differences between antibacterial-specific perturbations and the multitude of side effects related to cell death[19]. Accordingly, our analysis provides convincing evidence that the outer membrane, beyond its well-known role as a permeability barrier[52], is a key player in the mode of action of β-lactams and in L,D-transpeptidase-mediated β-lactam resistance, as previously found for tolerance to these drugs in various bacterial species[53–56]. In turn, our analysis identifies potential actionable targets to boost the efficacy of β-lactams against resistant bacteria.

## Methods

### Strains, plasmids, and growth conditions

The characteristics and origin of plasmids and strains used in the study are listed in Supplementary Table S1. Bacteria were grown at 37 °C in brain heart infusion (BHI; Difco) agar or broth with aeration (shaking @ 180 rpm). Kanamycin at 50 μg/ml was used for the selection of transductants carrying the Km^R cassette obtained from the Keio collection. The growth media were systemically supplemented with drugs to counter-select plasmid loss: 10 μg/ml tetracycline for plasmid pKT2, 20 μg/ml chloramphenicol for pKT8, 25 μg/ml zeocin for pHV30 and derivatives. Induction of the P_{trc}, P_{araBAD}, and P_{rhaBAD} promoters was performed with isopropyl β-D-1-thiogalactopyranoside (IPTG, 40 μM), L-arabinose (1%), and L-rhamnose (0.2%), respectively. Plasmids constructed in this study were obtained by using NEBuilder HiFi DNA assembly (New England Biolabs) method, unless otherwise specified. Deletions of specific genes were obtained by P1 transduction of the Km^R cassette of selected mutants from the Keio collection[57,58]. For multiple gene deletions, the Km^R cassette was removed by the FLT recombinase encoded by plasmid pCP20.

### Insertion library construction

A fresh colony of *Escherichia coli* BW25113 Δ*relA* pKT2(*ycbB*) pKT8(*relA'*) was grown in 10 ml of BHI broth supplemented with 10 μg/ml tetracycline and 20 μg/ml chloramphenicol. At an optical density at 600 nm (OD_{600nm}) of *ca*. 0.8, cells were harvested by centrifugation and washed four times with 10 ml H_2O at 4 °C. Multiple electroporations (see below) were performed using 100 μl of electrocompetent cells and 1 μl of EZ-Tn5 < KAN-2> transposome® (Lucigen). For each electroporation, transformed cells were incubated at 37 °C for 1 h in 2 ml of BHI broth supplemented with 10 μg/ml tetracycline and 20 μg/ml chloramphenicol. Cells were plated on 20 BHI agar plates (100 μl per plate) supplemented with 50 μg/ml kanamycin, 40 μM IPTG, and 1% L-arabinose in the absence (condition −CRO) or presence (condition +CRO) of 8 μg/ml ceftriaxone. Plates were incubated at 37 °C for 16 h (condition −CRO) or 24 h (condition +CRO). Cells were recovered from sets of 20 plates (corresponding to the same electroporation) in two steps by scrapping with 4 ml of BHI broth containing 20% glycerol followed by an additional wash of the plates with 4 ml of the same medium. The bacterial suspensions obtained for each electroporation were kept at −80 °C. For the −CRO condition, a total of 810,000 CFUs was obtained in 7 electroporations. For the +CRO condition, a total of 260,000 CFUs was obtained in 10 electroporations.

For DNA extraction, 10 μl of the bacterial suspensions obtained in the 7 (−CRO condition) or 10 (+CRO condition) electroporations were pooled. DNA extraction was performed with the Wizard Genomic DNA purification kit (Promega) in duplicate to obtain technical repeats.

### Junction fragments sequencing

The preparation of DNA libraries and the Illumina sequencing were performed by Viroscan3D company and the ProfileXpert platform of the Université de Lyon 1. Briefly, DNA was fragmented (*ca*. 300 bp) and P5 Illumina adapters were linked to the fragmented DNA. Junction fragments on both sides of EZ-Tn5 were amplified using standard P5 Illumina primer in association with Tn5-hybridizing primer HV1 or HV2 carrying the P7 Illumina adapter on the 5′ end (Supplementary Table S1). Indexes were carried by the Tn left or Tn right primers. Illumina sequencing was performed on a NextSeq Mid Output 150 bp (paired-end: 40–110). Demultiplexing and trimming of the reads were performed prior to the alignment with the *E. coli* BW25113 reference genome (NCBI accession number: CP009273).

## Tn-seq data analysis

Analysis was performed with the Python (v.3.7) package TRANSIT (v.3.0+)[59]. The TPP component was used to process raw experimental reads and generate WIG files for downstream analysis. Illumina paired ends reads with EZ-Tn5 insertion sites were identified by first selecting those with the expected 'TAAGAGACAG' transposon end sequence between nucleotides 25 and 50 of each read. The 110 bp reverse sequence (Adapter P7 sequencing primer) was used in all cases as it contained both EZ-Tn5 and genomic sequences. EZ-Tn5 reads were then mapped to the *E. coli* BW25113 reference sequence using BWA (v.0.7.17)[60]. Combined WIG files corresponding to left and right junctions were generated for each experiment. Transposon insertion events were assigned to specific open reading frames/genes using the GenBank (https://www.ncbi.nlm.nih.gov/assembly/) GFF3 file corresponding to the FASTA genomic sequence. Normalization of the number of inserts per ORF/gene was performed by Trimmed Total Reads (TTR): total read-counts normalized after trimming top and bottom 5% of read-counts. Mean insertion counts were calculated for each experimental condition: mean of left and right junctions from each of two experiments.

## Determination of gene essentiality and fitness cost

The junction fragments were aligned to the *E. coli* BW25113 reference genome and the essential genes were called by using the Tn5Gaps algorithm of the TRANSIT pipeline (Supplementary Data File 1a)[59]. The set of selectively essential gene in the +CRO condition were constructed by: (1) Identification of genes essential in the two replicates of the +CRO and not essential in the two replicates of the −CRO (subset 1). (2) Adding genes with one discrepancy among technical duplicates, e.g. marked as non-essential in one +CRO dataset or essential in one −CRO dataset (subset 2). (3) Average reads per gene were calculated to identify transposon insertions that are significantly (*p*-value < 0.05) underrepresented in the +CRO condition (Supplementary Data File 1a). Genes from subsets 1 & 2 that yielded a *p*-value > 0.05 or a fold-change < 4 were eliminated. The resulting set of 179 genes is listed in Supplementary Data File 1b. For pathways or gene clusters containing selectively essential genes in the +CRO condition, we also considered non-essential genes displaying a significantly reduced number of insertions in the +CRO condition (*p* < 0.05). These genes were deemed to incur a fitness cost.

## Plasmids and donor DNAs construction for CRISPR-Cas9 mutagenesis

The 20-nucleotide guiding sequences targeting *mrdA* (GTCTA-CAGTTAAACCCTATG) and *rodA* (GGTGGCTGCACAGCCTGACC) were independently introduced in plasmid pTargetF[61] by inverted PCR amplification using the primers HV3 and HV4 or HV5 and HV6 (Supplementary Table S1) to generate pTargetF-*mrdA* and pTargetF-*rodA*, respectively. Briefly, pTargetF was amplified by PCR with Phusion DNA polymerase (Thermo Scientific), the PCR products were digested by DpnI restriction enzyme (Thermo Scientific) and purified by agarose gel electrophoresis. Purified PCR products were phosphorylated and ligated with the T4 Polynucleotide Kinase and the T4 DNA ligase (Thermo Scientific) in a one pot reaction, and the resulting pTargetF-*mrdA* and pTargetF-*rodA* plasmids were introduced into *E. coli* TOP10 by electroporation.

The *mrdA* S[330]A, *mrdA* S[330]C, *rodA* D[159]A, *rodA* D[159]N, Δ*mrdA*, and Δ*rodA* donor DNAs were synthesized by GeneCust and cloned in pUC57. Donor DNAs used to introduce point mutations correspond to the sequence of the *mrdA* or *rodA* with missense mutation in the codon of interest and silent mutations in the sequence corresponding to the 20 nucleotides used to guide Cas9 on the chromosomic copy of the gene (to prevent subsequent cleavage by Cas9 after allelic exchange). Donor DNAs used to introduce deletions correspond to the 100 bp upstream, the first six codons, the last eight codons and the 100 bp

downstream sequences of *mrdA* or *rodA*. The donor DNAs were independently amplified by PCR with Phusion DNA polymerase (Thermo Scientific) and primers HV7 to HV14 depicted in Supplementary Table S1. PCR products were digested with DpnI restriction enzyme (Thermo Scientific) and purified by agarose gel electrophoresis.

## CRISPR-Cas9-mediated deletions and point mutations

Method adapted from Jiang Y. et al.[61]. A fresh colony of *E. coli* BW25113 Δ*relA* pKT8(*relA′*) pCas was grown at 30 °C with agitation in 10 ml of BHI broth supplemented with 1% L-arabinose to induce expression of both *relA′* and the λ Red system, and with 20 μg/ml chloramphenicol and 50 μg/ml kanamycin to counter-select loss of the plasmids. At an $OD_{600nm}$ of *ca.* 0.8, cells were harvested by centrifugation and washed four times with 10 ml $H_2O$ at 4 °C. Bacteria were electroporated with plasmid pTargetF-*mrdA* and the donor DNA *mrdA* S[330]A, *mrdA* S[330]C or Δ*mrdA*, or with plasmid pTargetF-*rodA* and the donor DNA *rodA* D[159]A, *rodA* D[159]N or Δ*rodA* with a molecular ratio of 1 plasmid for 5000 donor DNA molecules. Transformed cells were incubated at 30 °C for 2 h in 1 ml of BHI broth supplemented with 1% L-arabinose, 20 μg/ml chloramphenicol, and 50 μg/ml kanamycin. Bacteria were plated on BHI agar supplemented with 1% L-arabinose, 50 μg/ml kanamycin, and 120 μg/ml spectinomycin and incubated at 30 °C for 24 h. To get rid of pTargetF-*mrdA* or pTargetF-*rodA*, transformants were isolated on the same medium additionally supplemented with 500 μM IPTG for induction of the pCas-encoded sgRNA targeting the pTargetF plasmid and incubated at 30 °C for 24 h. The chromosomic deletions or point mutations were verified by PCR and Sanger sequencing. To get rid of the thermosensitive pCas plasmid, clones harboring the expected mutations were grown in BHI broth supplemented with 1% L-arabinose and 500 μM IPTG at 37 °C for 6 h and isolated on BHI agar supplemented with 1% L-arabinose at 37 °C for 16 h. Isolated clones were subcultured in BHI broth supplemented with 1% L-arabinose at 37 °C for 6 h and isolated on BHI agar supplemented with 1% L-arabinose at 37 °C for 16 h. The susceptibility of isolated clones to spectinomycin (loss of pTargetF derivatives) and kanamycin (loss of pCas) was tested to confirm the loss of these plasmids.

## Plating efficiency assay

Bacteria were grown to the late exponential phase, i.e. to an $OD_{600nm}$ 1.0 to 4.0 (*ca.* 6 h at 37 °C with vigorous shaking). The $OD_{600nm}$ was adjusted to 1.0 and 10-fold dilutions ($10^{-1}$ to $10^{-5}$) were prepared in BHI broth. Five μl of the resulting bacterial suspensions were spotted on BHI agar plates supplemented with inducers and drugs as indicated in the legend to figures. For the disk diffusion assay, 5 μl of the bacterial suspension at an $OD_{600nm}$ of 1.0 was inoculated in 5 ml of water. BHI agar plates were flooded with the latter suspension, excess liquid was removed, and the plates were kept at room temperature for 15 min prior to the addition of paper disks containing antibiotics. For the chlorophenol red-β-D-galactopyranoside (CPRG) assay, strain BW25113(*ycbB*, *relA′*) harboring pHV30bis(*lacZ*) was used since the parental BW25113 strain do not express the chromosomic copy of *lacZ*. Plates were imaged or the inhibition diameters were measured after 16 h (or 24 h for plates containing ceftriaxone) of incubation at 37 °C. Data shown in the figures are representatives of at least two biological repeats and the inhibition diameters measured are the median of three biological replicates.

## Preparation of sacculi

Bacteria were grown in BHI broth to the late exponential phase, i.e. to an $OD_{600nm}$ greater than 1.0 (*ca.* 6 h at 37 °C under agitation). Ten μl of the resulting bacterial suspensions were platted on BHI agar supplemented with inducers (40 μM IPTG and 1% L-arabinose) and drugs (10 μg/ml tetracycline and 20 μg/ml chloramphenicol, or 8 μg/ml ceftriaxone). For each replicate, 5 BHI agar plates were used in order to obtain a sufficient amount of bacteria. Plates were incubated for 16 h

(or 24 h for plates containing ceftriaxone) at 37 °C. Bacteria were harvested in two steps by scrapping each plate with 1 ml of phosphate-buffered saline (PBS) pH 7.2 followed by an additional wash of the plate with 1 ml of PBS. Bacteria were boiled in 0.5 × PBS supplemented with 4% sodium dodecyl sulfate (SDS) in a final volume of 20 ml for 1 h. Sacculi were harvested by centrifugation (20,000 × g for 20 min at 20 °C), washed five times with 20 ml of water, resuspended in 1 ml of 20 mM Tris-HCl pH 7.5, and incubated with 100 μg/ml pronase at 37 °C for 16 h. Sacculi were washed five times with 1 ml of water, resuspended in 1 ml of 20 mM sodium phosphate pH 8.0 and incubated with 100 μg/ml trypsin at 37 °C for 16 h. Sacculi were washed five times with 1 ml water, boiled for 5 min, collected by centrifugation, resuspended in 300 μl of water, and stored at −20 °C.

### Peptidoglycan analysis
Ten μl of purified sacculi were digested with 120 μM lysozyme in 40 mM Tris-HCl pH 8.0 at 37 °C for 16 h. Insoluble material was removed by centrifugation at 20,000 × g in a microcentrifuge for 10 min, and the soluble fraction containing muropeptides was reduced with sodium borohydride in 125 mM borate buffer pH 9.0 for 1 h at room temperature. Phosphoric acid was used to adjust the pH to 4.0. Muropeptides were separated by *rp*HPLC in a C18 column (Hypersil GOLD aQ; 250 × 4.6 mm; 3 μm, Thermo Scientific) at a flow rate of 1 ml/min with a linear gradient (0–20%) applied between 10 and 60 min (buffer A, TFA 0.1%; buffer B, acetonitrile 99.9%, TFA 0.1%, v/v). Absorbance was monitored at 205 nm and fractions were collected, lyophilized, resuspended in water and analyzed by mass spectrometry. Mass spectra were obtained on a Bruker Daltonics maXis high-resolution mass spectrometer (Bremen, Germany) operating in the positive mode (Analytical Platform of the Muséum National d'Histoire Naturelle, Paris, France). Mass spectral data were explored using mineXpert2[62].

### Western blot analysis
BW25113(*ycbB*, *relA'*) was grown in the −CRO and +CRO conditions in BHI agar plates as performed for the preparation of sacculi. Bacteria were lysed with 4% SDS at 96 °C for 45 min. Crude bacterial extracts were separated by SDS-PAGE. Lpp and RpoA (used as a loading control) were detected with polyclonal rabbit and mouse antibodies provided by J.F. Collet and C. Beloin, respectively. Primary antibodies were diluted at 1/10,000 (v/v). Detection was performed with peroxidase-coupled anti-rabbit (Sigma) and anti-mouse antibodies (Sigma) according to the manufacturer instructions of the Pierce™ ECL Western Blotting Substrate (Thermo Scientific). Secondary antibodies were diluted at 1/2000 (v/v).

### Lipid peroxidation detection
BW25113(*ycbB*, *relA'*) was grown at 37 °C in BHI broth supplemented with 40 μM IPTG and 1% L-arabinose to induce expression of *ycbB* and *relA'*, respectively. At an OD$_{600nm}$ of *ca*. 0.25 ceftriaxone was added and incubation was continued for 2 h. Bacteria (1 ml) were harvested, lysed, and the concentration of malondialdehyde (MDA) was determined according to the manufacturer instructions of the Lipid Peroxidation MDA Assay Kit (Sigma).

### Statistics and reproducibility
For Tn-seq analysis, reproducibility is addressed by the correlation of normalized average read numbers per CDS for two sequenced technical replicates obtained for the −CRO and +CRO conditions (Fig. 2a, b). The statistical treatment of the Tn-seq data is described under the heading "Determination of gene essentiality and fitness cost" in the "Methods" section. *p*-values were obtained from unpaired two-tailed *t*-tests. For the plating efficiency assays, at least two biological repeats were performed (data in figures originate

from one representative experiment). For peptidoglycan analysis by *rp*HPLC and mass spectrometry, three independent biological repeats were performed (data in figures originate from one representative experiment). Antibiotic susceptibility testing data are medians from three independent biological repeats. For Western blot analysis, images are one representative of three biological repeats. The concentrations of malondialdehyde (MDA) are the mean and standard deviation from three biological repeats. *p*-values were obtained from unpaired two-tailed *t*-tests.

### Reporting summary
Further information on research design is available in the Nature Portfolio Reporting Summary linked to this article.

## Data availability
The Tn-seq data generated in this study have been deposited in the Sequence Read Archive database (SRA) under accession code PRJNA907050. Source data are provided with this paper.

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

## Acknowledgements

This work was supported by the French National Research Agency ANR 'RegOPeps' (grant ANR-19-CE44-0007 to JEH). HV is the recipient of a doctoral fellowship from Sorbonne-Université (ED 515, Complexité du Vivant). We thank J. Lachuer and J. Bertrand of the ProfileXpert/Viroscan3D genomic platform for transposon insertion sequencing. We thank A. Marie for technical assistance in the collection of mass spectra at the Plateau Technique de Spectrométrie de Masse Bio-Organique of the Muséum national d'Histoire Naturelle. We thank J.F. Collet and C. Beloin for the generous gift of anti-Lpp and anti-RpoA antibodies, respectively. We thank Z. Edoo and F. Rusconi for critical reading of the manuscript.

## Author contributions

H.V.: conception and design, acquisition, analysis, and interpretation of data, drafting and revising the article. S.K.: analysis of data, drafting and revising the article. G.S.: drafting and revising the article. M.A. and J.E.H.: conception and design, analysis and interpretation of data, drafting and revising the article.

## Competing interests

The authors declare no competing interests.
