## [Peer Review File · Nature Communications]

REVIEWER COMMENTS

Reviewer #1 (Remarks to the Author):

The manuscript by Voedts et al makes elegant use of a mutant that becomes beta lactam resistant by replacing PBP-mediated PG crosslinking with LDT-catalyzed 3,3 crosslinks. Screening for genes that become essential when this mutant grows in the presence of a beta lactam, they uncover a role for RodA-mediated PG polymerization, precursor availability and OM integrity, among many other things, in the survival of this mutant. Overall, this is nicely done and I appreciate the careful and logical way they walk us through their TnSeq results. The results are highly significant as they allow us detailed and comprehensive insight into the poorly-understood mechanism of action of beta lactam antibiotics, and intrinsic resistance mechanisms. All experiments are carefully controlled and tested in logical ways. I only have minor comments on the manuscript.

- Line 126 and other places: Make sure you refer to the supplementary data whenever discussing TnSeq results to make sure the reader understands that the TnSeq results were validated (important due to the high number of false positives in TnSeq experiments)
- Line 132 – not sure I would say MreB controls the localization of elongasome -if anything, it is the other way around, i.e. PBP2 seems to recruit the other elongasome components, including MreB (PMID: 32077853)
- May be a personal style choice, but I would replace “unessential” with “non-essential”
- Line 214 – this sentence is confusing. Which redundant enzymes functions were essential? Please either expand this section, or perhaps delete it (does not contribute much to the overall flow)
- Line 251 – this is quite interesting: Your screen should be answered by the ECA_{gpl} polymerase (according to your logical reasoning through elimination, ECA_{gpl} is essential, so the polymerase should also be essential), so maybe you inadvertently already identified it? Also, did you test a Δ ECA_{lps} Δ ECA_{cyc} double mutant? Either of those ECA types might be necessary, but redundant. Lastly, might this also have something to do with C55 availability, rather than with the essentiality of ECA? This could be discussed.
- What is your model for beta lactams preventing LPP attachment? Might this be due to competition with enhanced PG 3,3 crosslink formation (sequestering LDT activity away from Braun’s)?
- Line 305 – I would not call this “elusive” anymore, given the wealth of literature on the subject, some of which you cite (PMID: 2875060, PMID: 25480295, PMID: 33274447 etc.). Maybe “poorly-understood” is better.
- Line 342 – Fig.1E is mentioned in reference to the essential role of regulators, but this figure shows PG fragment schematics. Also, did any oxidative stress response genes answer this screen, e.g. OxyR? This would lend more weight to the assessment in lines 340 – 343.

- Line 370 – the OM is not an “unsuspected” key player – there is plenty of literature on the essentiality/importance of the OM during beta lactam exposure in numerous bacteria (PMID: 23103254, PMID: 300612, PMID: 31285232, PMID: 24419348, PMID: 3563873, PMID: 35130322).

Reviewer #2 (Remarks to the Author):

In *E. coli*, the D,-D-transpeptidase activity of PBPs is essential for peptidoglycan (PG) synthesis and growth. *E. coli* is therefore sensitive to beta-lactams such as ceftriaxone (CRO) that inhibit PBPs, but this sensitivity can be suppressed by overproducing the YcbB L,-D transpeptidase and the ppGpp alarmone. In this study, Voedts et al. used a strain overproducing YcbB and ppGpp in a TnSeq screen to identify genes that are selectively required for survival in the presence of CRO. The authors identified 179 genes and focused this study on those affecting PG and outer membrane biogenesis. As stated in the Discussion section, the authors concluded that: 1) CRO prevented the “L,D-transpeptidase-mediated anchoring of the Braun lipoprotein to PG and increased outer membrane permeability”, 2) “the β -lactam-induced futile cycle was only sustainable if the efficacy of the peptidoglycan assembly pathway was not compromised by ineffective biosynthetic steps or by competition with the assembly of other polymers relying on the common undecaprenyl lipid transporter”, and 3) “the integrity of the cell envelope heavily relied on outer membrane polymers that were otherwise dispensable for growth.”

Overall, this extensive study was well done, and the data are of high quality. The manuscript is clearly written; the authors did a great job in explaining (and displaying in figures) several complex pathways. I also appreciate that they carefully tested a subset of their TnSeq hits so that they could convincingly show the requirement for robust PG synthesis and outer membrane biogenesis when their strain was exposed to the beta-lactam. However, these two main conclusions are not surprising since, as the authors point out, it is known that both PG and the outer membrane contribute to the structural integrity of the cell and prevent osmotic lysis. Furthermore, the authors pitch this study as if their findings reveal something specific about the effect of beta-lactams. However, it is unclear if similar results would have obtained if PG biosynthesis had been decreased in a different way or if some or most of the results are specific to the strain used, which overproduces YcbB and ppGpp. Lastly, the conclusion that CRO prevents the crosslinking of Lpp to PG needs further support. Together, these issues limit the impact of this study with respect to advancing our understanding of the effect of beta-lactams on envelope biogenesis, but I think it would be a very useful resource for the community.

Specific major points:

1) As stated above, it is difficult to know if the effects that the authors report are specific to the treatment with beta-lactams or if similar results would be obtained in strains with weakened PG

regardless of the reason for the defects. The authors should at least recognize this issue in the manuscript. Moreover, although I appreciate the strategy of using a strain that can grow in the presence of CRO, it is difficult to know if the insertions identified have an effect because the respective genes are required to specifically cope with the effects of CRO or because the insertions interfere (directly or indirectly) with the suppression by YcbB and/or ppGpp.

2) In several figures, the spot assays show that the strain overproducing YcbB and ppGpp grows mucoid in the presence of CRO, implying that the Rcs envelope stress response is activated. This is not surprising since Rcs can be activated by PG-targeting drugs such as mecillinam. In *E. coli*, very high levels of Rcs activation can be lethal. For example, the Rcs inhibitor regulator IgaA is essential unless genes encoding downstream Rcs regulators are deleted. Given that there are multiple ways to turn on Rcs and the nature of the activating signals are poorly understood, it is possible that some of the insertions in genes that are selectively essential could be inducing the Rcs response as well. This possibility calls into question whether hyperactivation of Rcs could be the cause of death (in a non-specific manner). In fact, Fig. 3D/Fig. S2 shows that mutants with insertions in PG biosynthesis genes are mucoid in the absence of CRO. Does removing Rcs rescue any of the mutants/reverse the essentiality of pathways? Certainly, the lethality of the capsule pathway mutants would be expected to be reversed. Insertions that activate Rcs would drive undecaprenyl to the capsule biosynthetic pathway, potentially limiting the amount of undecaprenyl available for PG biogenesis, a situation that the authors have shown to be lethal when their strain is treated with CRO.

3) The authors concluded that Lpp crosslinking to PG is inhibited by CRO treatment in their strain. Have the authors checked Lpp protein levels? There was large decrease in the levels of Lpp crosslinks, but the authors need to rule out that this decrease is not caused by a decrease of Lpp protein levels caused by the treatment.

4) Discussion, lines 339-343: The authors state that “elevated anabolic demand implies that ROS production remains elevated. The successful defense against the resulting oxidative stress may account for the selectively essential roles of regulators, chaperones, and enzymes involved in the repair of macromolecules.” This statement is highly speculative and needs further details since it is relevant to beta-lactam sensitivity. Do the authors have any evidence that ROS production is elevated? Is there a clue from TnSeq results that this is the case? Are any of the genes known to be involved in combating ROS identified?

5) As I mentioned above, I do not think that this study significantly advances our understanding of PG biosynthesis or the effect of or resistance to b-lactams. There are two possible areas that might increase the impact of the study (pending findings): a) insertions that increased fitness, which were not mentioned in this study; and b) whether any of the insertions increase sensitivity to CRO on their own. The authors speculate about the latter point at the end of the Discussion by stating that their screen

identified genes and responses that prevent bacterial killing by β -lactams that could be “actionable targets to boost the efficacy of β -lactams against resistant bacteria.”

Minor points:

1) Line 86 should state that ceftriaxone is a beta-lactam.

2) The statement about the dispensability of DapF needs a citation in line 199. In fact, the Results section needs citations when proteins and their functions are stated. Please revise.

3) Line 223: "lipid transporter" should be "lipid carrier" since I think they are referring to undecaprenyl.

4) Lines 259-260: "Additions a ..." should be "additions of a".

5) Fig. 6C: I assume the "Loading (ug)" label refers to the amount of drugs loaded onto disks. Please clarify in legend.

Reviewer #3 (Remarks to the Author):

This is an interesting and potentially transformative paper on the basis of bacterial cell wall peptidoglycan formation and the relationship that has to beta-lactam antibiotic mode of action. Moreover, they appear to establish a functional linkage between the RodA dependent elongasome machinery and outer membrane biosynthesis which is entirely novel. I am very supportive of this paper's eventual publication in Nature Communications and believe it could be a transformative paper. The authors present some very interesting and impactful data generally and the figures in particular, are excellent quality.

However, I am critical of the manuscript with respect to its ability to relate to a wider scientific audience consistent with Nature communications rather than a specialist microbiology community which at present, this paper is clearly designed for. Some additional explanation is required in the text to ensure that the paper is accessible to a wide audience who may not be familiar with this research group's impressive and detailed track record in this field of microbiology. This is required in some detail to understand the context of what is written here. A good exemplar of this deficiency is the central experimental approach of growth in the presence or absence of ceftriaxone with the transposon library

created. There is absolutely no explanation as to why that antibiotic is chosen or has relevance in this context!

I trust the authors will find my comments helpful in the development of their paper as they are intended to be constructive criticism

Abstract

Line 14 "bypass" this is not a useful term in the context of the abstract. One presumes that authors mean "functional replacement" of PBPs by LD-transpeptidases to enable peptidoglycan biosynthesis. Whilst I appreciate the authors have used this term before in the literature some additional context/explanation is required to improve the accessibility of the paper. I also think that the three sentences from line 15 to 21 that summarise the major results of the paper are somewhat disjointed and do not naturally produce a narrative that endorses the final sentence of the abstract. In short, they authors need a better abstract and this a weakness of the paper as present.

Introduction.

Line 34 I Suggest that the word subsequently is inserted after "chains that are"

Line 36. The word "cleave is misleading and technically incorrect. The terminal D-ala of the pentapeptide is not removed as implied as some form of catabolic process but as a consequence of the mechanism of formation of a PEP-peptidoglycan intermediate that can be broken down in a number of ways.

Line 37. Do we need and arrow between 4-3?

Line 42. We have the word "Bypass" again but this time with a hyphen which is inconsistent. Whilst reference 8 is used to guide the reader, I would argue that their paper PLoS One 2013 Jul 4;8(7):e67831 actually provides a more useful and earlier description of "bypass"

Line 42 "unrelated" in what way? I don't believe this term is helpful to the reader, comprehension of science and is potentially misleading.

Line 58. "...for the rescue of PBPs by the 3-3...". The subject is peptidoglycan biosynthesis which is rescued by the LD-transpeptidases. The function of PBPs might be rescued in this context but not the PBPs since they are rendered useless by the beta lactam drugs.

Line 62. “Evolvability” ie capacity of a system for adaptive evolution. “is this what the authors actually mean?”

Results

Line 85. The authors need to provide some more background with a simple sentence on the strain in question. Inspection of Table S1 reveal nothing about this strain and reference 23 appears to be a methods paper not related to the strain.

Is the strain used BW25513(ycbB, relA') as in table S1? (Not BW15113 as stated..?)

Line 86 Similarly the Authors should provide a clear rational why ceftriaxone is used in this study and its relevance to PBP and LD-TP inhibitions as it is not apparent to a non-expert in beta-lactam development.

Line 86 Is a central tenet of the experiment that over expression of YcbB should overcome the inhibition of this enzyme by ceftriaxone produced from the chromosomal gene copy? If so, please make this clear.

Line 119-121. Many of the Fts genes listed are essential under any condition, so to try and describe this in the context of + and - antibiotic is misleading. Whatever the authors are trying to say it needs rewording. The last sentence from line 122 is a profound finding however!

lines 139-142. The authors state that ceftriaxone will inactivate PBP2 and cite their paper (Ref 8) and that of Erin Carlson (ref 32). The latter suggests that the primary affinity target of ceftriaxone by at least 20 fold is PBP3 in E.coli not PBP2. Thus, this statement needs to be held in context of the concentration of antibiotic used in the experiment. Can they say categorically that PBP2 is inactivated under these conditions? If so, what of YcbB? Ceftriaxone also inhibits this enzyme I believe and is a central tenet of their paper from 2013: PLoS One 2013 Jul 4;8(7):e67831.doi: 10.1371/journal.pone.0067831.

Line 199. We need a reference to the dispensability of dapF at the end of the sentence that describes it.

Line 201-210. The authors postulate that PBPs are able to cross link stem peptides form with LL-DAP instead of meso-(DL)-DAP and provide reference to examples in the literature that support LL-DAP dependent growth. Patin et al <https://pubmed.ncbi.nlm.nih.gov/20659527/> report that the Km of Meso-DAP for MurE is 40mM. Thus the 3000-fold difference in concentration to allow LL-DAP incorporation postulated in the manuscript here is 120mM. This seems unlikely is in 10 fold higher than that reported by Mengin-Lecreulx et la J Bacteriol 1988 May;170(5):2031-9 who also discuss “The low extent of cross

linkage of the peptidoglycan isolated from the *dapF* mutant suggested that the presence of an LL-DAP residue interfered in some way or another with this late step of peptidoglycan biosynthesis". Moreover whilst LL-DAP incorporation into *E. coli* peptidoglycan is possible in a laboratory setting it is clear that the preferred substrate for peptidoglycan formation is predominantly Meso-DAP.

Cathewood et al 2020 (<https://pubmed.ncbi.nlm.nih.gov/32048840/>) provides an alternative explanation in that in the presence of LL-DAP as a donor in the transpeptidase reaction *E. coli* PBP1b switches to formation of DD-carboxypeptidase activity predominantly, i.e. the formation of tetrapeptides which are the required substrates for LD-transpeptidases including YcbB. If this stereochemical bias is also found in other class A and class B PBPs then this may account for the observations seen. It would be interesting to know if the crosslinking composition of PG from cells lacking *DapF*.

Line 216-221-If "all but one (of the) colonic acid biosynthetic enzymes, *Wca* were (are) specific essential in the +CRO conditions", how is the next sentence correct?

This entire section from line 216 is difficult to follow and understand the logic of.

Line 242. Again, this section appears hard to reconcile and depends how the experiment has been conducted. The lack of essentiality in individual mutations by TRADIS in the genes responsible for ECA polymer formation suggests functional redundancy between the three ECA polymer systems surely. The result that many of these genes become essential under the +CRO condition indicates a connection between LD transpeptidases and the PBPs with outer membrane integrity

Line 272-282. The logic in this section is difficult to follow and a prerequisite appears to be an encyclopaedic knowledge of peptidoglycan chemistry and its relationship to LD transpeptidase required for Braun's lipoprotein anchoring (*LdtA-C*) as opposed to 3-3 crosslink formation (*LdtD*, *LdtE*). The fact that *LdtD* and *YcbB* are the same is not helpful. None of this is explained to enable the reader to understand the argument being made and needs to be rewritten to provide that basis.

Lines 283-300

Arguably the best written section of this latter part of the manuscript. It would have been perfect with just one additional sentence that made clear that the CPRG assay assesses outer membrane integrity by detection of LacZ enzyme activity in the media as a result of leakage (or words to that effect).

Lines 318-329

Whilst the concluding sentence of this paragraph appears to be supported by the paper, the preceding lines require an expert level appreciation of the most recent literature in order to interpret. Also do the

authors actually mean what is said regarding pre-synthesis of 3-3 crosslinked strands which are then inserted into preexisting PG. Ho is this mediated, further 3-3 crosslinking? This will not do.

Point by point answers to reviewers' comments

Reviewer #1:

The manuscript by Voedts et al makes elegant use of a mutant that becomes beta lactam resistant by replacing PBP-mediated PG crosslinking with LDT-catalyzed 3,3 crosslinks. Screening for genes that become essential when this mutant grows in the presence of a beta lactam, they uncover a role for RodA-mediated PG polymerization, precursor availability and OM integrity, among many other things, in the survival of this mutant. Overall, this is nicely done and I appreciate the careful and logical way they walk us through their TnSeq results. The results are highly significant as they allow us detailed and comprehensive insight into the poorly-understood mechanism of action of beta lactam antibiotics, and intrinsic resistance mechanisms. All experiments are carefully controlled and tested in logical ways. I only have minor comments on the manuscript.

- *Line 126 and other places: Make sure you refer to the supplementary data whenever discussing TnSeq results to make sure the reader understands that the TnSeq results were validated (important due to the high number of false positives in TnSeq experiments)*

We have cited the Supplementary Data files lines 138, 193, 220, 240, 267, and 284 as requested.

- *Line 132 – not sure I would say MreB controls the localization of elongasome -if anything, it is the other way around, i.e. PBP2 seems to recruit the other elongasome components, including MreB (PMID: 32077853)*

We changed the sentence from "... MreB, an actin-like protein forming a cytoskeleton that controls the localization and movement of the elongasome complex⁶" to "... MreB, an actin-like protein forming a scaffolding cytoskeleton for the elongasome complex⁶".

- *May be a personal style choice, but I would replace "unessential" with "non-essential".*

As suggested by the reviewer, the word "unessential" was replaced by "non-essential" in the entire manuscript.

- *Line 214 – this sentence is confusing. Which redundant enzymes functions were essential? Please either expand this section, or perhaps delete it (does not contribute much to the overall flow)*

The sentence "The essentiality of genes encoding redundant enzyme functions was not expected and is reported in Supplementary Fig. S5." was changed to "Data for essential biosynthetic steps involving redundant enzyme functions are reported in Supplementary Fig. S5".

- **Line 251 – this is quite interesting: Your screen should be answered by the ECA_{agl} polymerase (according to your logical reasoning through elimination, ECA_{agl} is essential, so the polymerase should also be essential), so maybe you inadvertently already identified it? (i) Also, did you test a $\Delta ECA_{lps} \Delta ECA_{cyc}$ double mutant? (ii) Either of those ECA types might be necessary, but redundant. Lastly, might this also have something to do with C55 availability, rather than with the essentiality of ECA? This could be discussed.**

Answers to specific points raised by the reviewer are as follows:

- (i) There are 14 selectively essential genes of unknown functions and, as the reviewer pointed out, the ECA_{PGL} polymerase should be one of them. However, our analysis did not identify which of the 14 genes encodes this polymerase.
- (ii) The $\Delta ECA_{LPS} \Delta ECA_{CYC}$ double mutant ($\Delta waaL \Delta wzzE$) was constructed and was resistant to ceftriaxone (Supplementary Fig. S6b). Thus, ECA_{PGL} is the ECA required for resistance and this rules out the possibility that “Either of those ECA types might be necessary, but redundant.”
- (iii) The first gene encoding the first omitted step of ECA synthesis was essential (*wecA*; Fig. 5b and Supplementary Fig. S6b). Thus essentiality did not depend upon C55 availability, as shown for colanic acids, but on the specific requirement of ECA_{PGL} in the outer membrane.

- **What is your model for beta lactams preventing LPP attachment? Might this be due to competition with enhanced PG 3,3 crosslink formation (sequestering LDT activity away from Braun’s)?**

As described in our answer to comment #3 (reviewer 2), we have performed additional Western blot analyses showing that Lpp is produced in the +CRO condition. Thus, ceftriaxone prevents Lpp anchoring rather than Lpp production. The underlying mechanism is unlikely to involve competition with 3→3 peptidoglycan cross-linking by YcbB. Indeed, in the absence of ceftriaxone, Lpp anchoring was not affected despite a 40% contribution of YcbB to peptidoglycan cross-linking (Supplementary Fig. S7). We have purified the three LDTs responsible for Lpp anchoring and shown that none of these enzymes is effectively inhibited by ceftriaxone *in vitro*. The effect of ceftriaxone on Lpp anchoring is therefore unlikely to result from direct inhibition of these enzymes. In spite of these additional data, we have not identified the mechanism responsible for reduced Lpp anchoring in the +CRO condition. It may involve an indirect modulation of the activity of these enzymes in response to inhibition of PBPs by ceftriaxone. It may also involve enhanced activity of the YafK hydrolase that detaches the Lpp. Directly probing the *in vivo* activity of these enzymes is challenging and the corresponding assays remain to be developed.

- **Line 305 – I would not call this “elusive” anymore, given the wealth of literature on the subject, some of which you cite (PMID: 2875060, PMID: 25480295, PMID: 33274447 etc.). Maybe “poorly-understood” is better.**

As requested by the reviewer, “elusive” was replaced by “poorly-understood”.

➤ *Line 342 – Fig.1E is mentioned in reference to the essential role of regulators, but this figure shows PG fragment schematics. Also, did any oxidative stress response genes answer this screen, e.g. OxyR? This would lend more weight to the assessment in lines 340 – 343.*

(i) Line 342 should have cited “Fig. 2e, Supplementary Data file 1b” rather than “Fig. 1e, Supplementary Table S1”. We apologize for this mistake.

(ii) Several oxidative stress response genes were detected as detailed in our answer to major comment #4 from reviewer #2. However, *oxyR* itself was dispensable for growth both in the -CRO and +CRO conditions.

➤ *Line 370 – the OM is not an “unsuspected” key player – there is plenty of literature on the essentiality/importance of the OM during beta lactam exposure in numerous bacteria (PMID: 23103254, PMID: 300612, PMID: 31285232, PMID: 24419348, PMID: 3563873, PMID: 35130322).*

We agree that the word “unsuspected” was inappropriate with respect to previous publications addressing the role of the outer membrane in β -lactam tolerance mechanisms in various bacterial species. Consequently, we have modified the sentence appearing Line 370 (now 425) and cited a subset of the references mentioned by the reviewer:

The sentence:

“Accordingly, our analysis provides convincing evidence that the outer membrane, beyond its well-known role as a permeability barrier⁴⁸, is an unsuspected key player in the mode of action of β -lactams and in the mechanism of resistance to these drugs.”

Was changed into:

“Accordingly, our analysis provides convincing evidence that the outer membrane, beyond its well-known role as a permeability barrier⁵², is a key player in the mode of action of β -lactams and in L,D-transpeptidase-mediated β -lactam resistance, as previously found for tolerance to these drugs in various bacterial species⁵³⁻⁵⁶”

Reviewer #2

In E. coli, the D,D-transpeptidase activity of PBPs is essential for peptidoglycan (PG) synthesis and growth. E. coli is therefore sensitive to beta-lactams such as ceftriaxone (CRO) that inhibit PBPs, but this sensitivity can be suppressed by overproducing the YcbB L,D-transpeptidase and the ppGpp alarmone. In this study, Voedts et al. used a strain overproducing YcbB and ppGpp in a TnSeq screen to identify genes that are selectively required for survival in the presence of CRO. The authors identified 179 genes and focused this study on those affecting PG and outer membrane biogenesis. As stated in the Discussion section, the authors concluded that: 1) CRO prevented the “L,D-transpeptidase-mediated anchoring of the Braun lipoprotein to PG and increased outer membrane permeability”, 2) “the β -lactam-induced futile cycle was only sustainable if the efficacy of the peptidoglycan assembly pathway was not compromised by ineffective biosynthetic steps or by competition

with the assembly of other polymers relying on the common undecaprenyl lipid transporter”, and 3) “the integrity of the cell envelope heavily relied on outer membrane polymers that were otherwise dispensable for growth.”

Overall, this extensive study was well done, and the data are of high quality. The manuscript is clearly written; the authors did a great job in explaining (and displaying in figures) several complex pathways. I also appreciate that they carefully tested a subset of their TnSeq hits so that they could convincingly show the requirement for robust PG synthesis and outer membrane biogenesis when their strain was exposed to the beta-lactam. However, these two main conclusions are not surprising since, as the authors point out, it is known that both PG and the outer membrane contribute to the structural integrity of the cell and prevent osmotic lysis. Furthermore, the authors pitch this study as if their findings reveal something specific about the effect of beta-lactams. However, it is unclear if similar results would have obtained if PG biosynthesis had been decreased in a different way or if some or most of the results are specific to the strain used, which overproduces YcbB and ppGpp. Lastly, the conclusion that CRO prevents the crosslinking of Lpp to PG needs further support. Together, these issues limit the impact of this study with respect to advancing our understanding of the effect of beta-lactams or envelope biogenesis, but I think it would be a very useful resource for the community.

The negative evaluation of the novelty of our findings does not take into account the fact that the Tn-seq analysis was performed in conditions supporting growth in the presence of β -lactams, rather than enabling survival, as stated by the reviewer. The concentration of the β -lactam ceftriaxone used in the +CRO condition was sufficient to fully inactivate all transpeptidases belonging to the PBP family. This enabled us to study the consequences of PBP inactivation in actively dividing bacteria that produce a functional and osmoprotective cell envelope. In contrast, previous studies addressed the impact of β -lactams on bacteria that were exposed to bacteriostatic drug concentrations or to drug concentrations that resulted in lytic or non-lytic cell death in a few generations. The limitations of these previous experimental designs were recognized by the corresponding investigators, as exemplified by the following example: “The experimental measurement of metabolomic responses to antibiotics in bacteria requires trade-off decisions. Here, we chose to take metabolic measurements in nonlethal (bacteriostatic) media conditions, which provided greater confidence that changes in metabolite levels are the direct result of enzymatic inactivation of PBP2 rather than nonspecific by-products of cell death. In this context, it is possible that metabolic shifts measured in bacteriostatic media conditions may not translate precisely to the lethal (bactericidal) media conditions.” from reference 19 of the revised manuscript. In addition, we would also like to underscore that our study is original since it provides data on the inhibition of all PBPs by broad-spectrum ceftriaxone whereas most previous studies focused on selective inhibition of PBP2 by mecillinam.

The fact that our study concerns the impact of β -lactams on actively growing bacteria in spite of full PBP inactivation implies that our study design cannot be extended to other drug families or to other means of reducing peptidoglycan synthesis. Indeed, the crucial bypass resistance

mechanism mediated by YcbB is specific for β -lactams. Testing other means to reduce peptidoglycan synthesis is therefore out of the scope of the current study.

We acknowledge the fact that the role of the outer membrane in membrane stiffness has been previously reported, as indicated in the first version of the manuscript (lines 346-354 and cited references 46-47). However, the originality of our analysis stems from addressing the role of cell wall polymers in actively dividing cells, as indicated above. We also advocate that our study provides an unprecedented picture of the complexity of the interplay between cell envelope polymers since it is based on a genome-wide approach.

At variance with the reviewer's general evaluation of our work, we persist in concluding that our findings reveal something specific about the effect of beta-lactams since we report two new aspects of the mode of action of β -lactams: (i) A self-promoted increase in the permeability of the membrane, which obviously has major consequences for evaluating drug efficacy, as investigated for aminoglycosides. To the best of our knowledge, self-promoted penetration has not been previously reported for β -lactams. (ii) The inhibition of the anchoring of the Braun lipoprotein (Lpp) to peptidoglycan (PG) by β -lactams is also a novel finding. These two original aspects were not recognized by the reviewer probably because the concluding paragraph of the discussion was too condensed. We have rephrased this section of the manuscript:

“In conclusion, the rescue of PBPs by YcbB, coupled to a genome-wide Tn-seq screen, revealed three unexpected impacts of β -lactams. First, exposure to ceftriaxone conditionally prevented L,D-transpeptidase-mediated anchoring of the Braun lipoprotein to PG and increased outer membrane permeability (Supplementary Fig. S7; Fig. 6), which is, in itself, a previously unrecognized mode of action of β -lactams.”

Was changed into:

“In conclusion, the rescue of PBPs by YcbB, coupled to a genome-wide Tn-seq screen, revealed two unexpected impacts of β -lactams. First, exposure to ceftriaxone prevented L,D-transpeptidase-mediated anchoring of the Braun lipoprotein to PG (Supplementary Fig. S7). Second, exposure to ceftriaxone increased the permeability of the outer membrane (Fig. 6). The latter effect implies that self-promoted penetration through the outer membrane is a previously unrecognized facet of the mode of action of β -lactams. In addition” The other points were not modified.

The reviewer states that the conclusion that CRO prevents the crosslinking of the Braun lipoprotein (Lpp) to PG needs further support. We acknowledge the fact that the first version of the manuscript was only based on mucopeptide analysis. Additional analyses were performed to address this limitation. In the revised version of the manuscript, we added data showing that ceftriaxone specifically prevents Lpp anchoring as this protein was produced in the +CRO condition. This was established by Western blot analyses of crude bacterial extracts (see additional panel g of Supplementary Fig. S7).

Specific major points:

1) As stated above, it is difficult to know if the effects that the authors report are specific to the treatment with beta-lactams or if similar results would be obtained in strains with weakened PG regardless of the reason for the defects. The authors should at least recognize this issue in the manuscript. Moreover, although I appreciate the strategy of using a strain that can grow in the presence of CRO, it is difficult to know if the insertions identified have an effect because the respective genes are required to specifically cope with the effects of CRO or because the insertions interfere (directly or indirectly) with the suppression by YcbB and/or ppGpp.

We specifically addressed this issue by analyzing the peptidoglycan structure of mutants susceptible to ceftriaxone defective in the synthesis of membrane polymers ECA ($\Delta wecA$), colonic acids ($\Delta wcaI$), LPS core sugars ($\Delta waaC$), and the mutant defective in the production of the cytoplasmic L,D-carboxypeptidase ($\Delta ldcA$). This analysis revealed that defects in these polymers or in the L,D-carboxypeptidase production did not impair synthesis of 3→3 cross-linked dimers (see the additional muropeptide analyses appearing in the Table below). Thus, defects in these polymers did not interfere with peptidoglycan cross-linking by YcbB and (p)ppGpp. We prefer not including these additional controls in the revised version since this would complicate our manuscript.

Muropeptide composition of derivatives of BW25113(ycbB , relA') harboring the indicated deletions								
Muropeptide	$\Delta wecA$ (ECA _{PG})		$\Delta wcaI$ (colanic acid)		$\Delta waaC$ (LPS core sugars)		$\Delta ldcA$ (L,D-carboxypeptidase)	
	Repeat 1	Repeat 2	Repeat 1	Repeat 2	Repeat 1	Repeat 2	Repeat 1	Repeat 2
Tri	18	16	21	19	20	21	20	20
Tetra	18	22	21	23	22	24	23	22
3→3	38	37	24	24	27	23	23	23
4→3	23	22	31	31	29	30	30	30
Tri-KR	3	3	4	3	2	3	3	5

2) In several figures, the spot assays show that the strain overproducing YcbB and ppGpp grows mucoïd in the presence of CRO, implying that the Rcs envelope stress response is activated. This is not surprising since Rcs can be activated by PG-targeting drugs such as mecillinam. In E. coli, very high levels of Rcs activation can be lethal. For example, the Rcs inhibitor regulator IgaA is essential unless genes encoding downstream Rcs regulators are deleted. Given that there are multiple ways to turn on Rcs and the nature of the activating signals are poorly understood, it is possible that some of the insertions in genes that are selectively essential could be inducing the Rcs response as well. This possibility calls into question whether hyperactivation of Rcs could be the cause of death (in a non-specific manner). In fact, Fig. 3D/ Fig. S2 shows that mutants with insertions in PG biosynthesis genes are mucoïd in the absence of CRO. Does removing Rcs rescue any of the mutants/reverse the essentiality of pathways? Certainly, the lethality of the capsule pathway mutants would be expected to be reversed. Insertions that activate Rcs would drive undecaprenyl to the capsule biosynthetic pathway, potentially limiting the amount of

undecaprenyl available for PG biogenesis, a situation that the authors have shown to be lethal when their strain is treated with CRO.

We acknowledge that activation of capsule synthesis by Rcs could indirectly be responsible for the growth defect of mutants deficient in ECA_{PGL}, LPS core sugars, LdcA, or DapF production. This possibility was not addressed in the first version of the manuscript. We therefore constructed additional mutants combining inactivation of WcaJ (the first enzyme in the colanic acid synthesis pathway) to mutations affecting ECA, LPS core sugars, LdcA, or DapF production. As shown in Supplementary Fig. S6c and introduced Lines 290-297, deletion of genes encoding WcaJ did not abolish the requirement for ECA_{PGL}, LPS core sugars, LdcA, and DapF for YcbB-mediated β -lactam resistance. These additional data ruled out the possibility of an indirect and non-specific effect of capsule overproduction.

3) The authors concluded that Lpp crosslinking to PG is inhibited by CRO treatment in their strain. Have the authors checked Lpp protein levels? There was large decrease in the levels of Lpp crosslinks, but the authors need to rule out that this decrease is not caused by a decrease of Lpp protein levels caused by the treatment.

Additional Western blot experiments using polyclonal antibodies raised against Lpp showed that the overall Braun lipoprotein content of the envelope was not affected (see additional panel g in the revised Supplementary Fig. S7). Thus, the reduced abundance of muropeptide 4 (a disaccharide-tripeptide substituted by the Lys-Arg motif from Lpp, as defined in Fig. 1) was due to a defect in Lpp anchoring but not to a decrease in the production of this lipoprotein. These new data were introduced Lines 313 of the revised manuscript.

4) Discussion, lines 339-343: The authors state that “elevated anabolic demand implies that ROS production remains elevated. The successful defense against the resulting oxidative stress may account for the selectively essential roles of regulators, chaperones, and enzymes involved in the repair of macromolecules.” This statement is highly speculative and needs further details since it is relevant to beta-lactam sensitivity. Do the authors have any evidence that ROS production is elevated? Is there a clue from TnSeq results that this is the case? Are any of the genes known to be involved in combating ROS identified?

We are able to positively answer all three questions as follows:

(i) We have performed additional experiments providing evidence for ROS production by determining the oxidation of lipids.

The new data have been introduced at the end of the results section and in a new panel h of Supplementary Fig. S7:

“Damage to the outer membrane mediated by reactive oxygen species may be involved in this process since increased lipid peroxidation was detected by a colorimetric assay (Supplementary Fig. S7h).” Lines 350-352

In the Discussion section, we have mentioned that our claim for elevated ROS production is supported by determination of the peroxidation of lipids:

“In agreement, increased lipid peroxidation was detected in the +CRO condition (Supplementary Fig. S7h).” Lines 368-369

(ii) The Tn-seq results provided a clue that this is the case since the *trxB* gene encoding thioredoxin reductase was selectively essential in the +CRO condition. In addition, Tn insertions in the *gor* gene encoding glutathione reductase incurred a fitness cost in presence of ceftriaxone.

(iii) Several genes known to be involved in combating redox-mediated damages to proteins, DNA, and Fe-S clusters were identified as selectively essential in the +CRO condition (Supplementary Data File 1b) :

- For protein damages, relevant genes included periplasmic chaperones and proteases encoded by *surA*, *degP*, *skp*, *ppiD*, and *bepA*, and cytoplasmic chaperones encoded by *tig* and *ybbN*.
- For DNA repair, relevant genes included *polA*, encoding the DNA polymerase I involved in several DNA repair pathways, *dinG*, encoding a helicase involved in DNA repair, *priA* and *priC*, encoding proteins enabling blocked replication forks to resume, *mfd*, involved in transcription-coupled DNA repair, *ruvA* and *ruvB*, involved in the resolution of Holliday junctions.
- For assembly of Fe-S clusters, relevant genes included *iscU* and *iscA* encoding the Fe-S cluster scaffold and insertion proteins, respectively, as well as the associated chaperones encoded by *hscA* and *hscB*.

5) As I mentioned above, I do not think that this study significantly advances our understanding of PG biosynthesis or the effect of or resistance to β -lactams. There are two possible areas that might increase the impact of the study (pending findings): a) insertions that increased fitness, which were not mentioned in this study; and b) whether any of the insertions increase sensitivity to CRO on their own. The authors speculate about the latter point at the end of the Discussion by stating that their screen identified genes and responses that prevent bacterial killing by β -lactams that could be “actionable targets to boost the efficacy of β -lactams against resistant bacteria.”

a) Insertions that increased fitness concerned a limited number of genes as shown by the volcano plot appearing in Fig. 2d (five genes in the upper right sector). Neither the fold-change nor the significance of these five genes are impressive. These genes are *rapA*, encoding the RNA polymerase recycling factor, *proQ*, encoding a RNA chaperone protein, *rbfA*, encoding a 30S ribosome binding factor, *hemC*, encoding a protein involved in tRNA-dependent tetrapyrrole synthesis, and *tusB*, encoding a protein involved in tRNA 2-thiouridine synthesis. The identity of these five proteins suggests that they might affect transcription and translation. Thus, they are not directly related to the functions discussed in the current study (peptidoglycan synthesis, envelope polymers, redox stress response) but could be related to the participation of

(p)ppGpp in resistance. We therefore think that discussing these functions is out of the scope of the current manuscript.

b) We refer to actionable targets to boost the efficacy of β -lactams against resistant bacteria (as opposed to susceptible bacteria). Therefore, the statement does not refer to insertions that might increase sensitivity to ceftriaxone on their own. To satisfy the curiosity of the reviewer, we determined the susceptibility of *ldcA*, *waaC*, and *wcaI* mutants to ceftriaxone by the disk diffusion assay and did not detect important modifications (≤ 3 mm) of the inhibition zones (35 to 38 mm). It therefore appears that genes such as *ldcA*, *waaC*, and *wcaI* are potential actionable targets against resistant bacteria, as indicated in the discussion, but not against susceptible bacteria.

Minor points:

1) Line 86 should state that ceftriaxone is a beta-lactam.

We indicated in Lines 94-98 that ceftriaxone is a β -lactam as requested.

2) The statement about the dispensability of DapF needs a citation in line 199. In fact, the Results section needs citations when proteins and their functions are stated. Please revise.

References for protein functions was added lines 199 as requested.

3) Line 223: "lipid transporter" should be "lipid carrier" since I think they are referring to undecaprenyl.

The text was modified as requested.

4) Lines 259-260: "Additions a ..." should be "additions of a".

The text was modified as requested.

5) Fig. 6C: I assume the "Loading (ug)" label refers to the amount of drugs loaded onto disks. Please clarify in legend.

Fig. 6c has been clarified as requested.

Reviewer #3

This is an interesting a potentially transformative paper on the basis of bacterial cell wall peptidoglycan formation and the relationship that has to beta-lactam antibiotic mode of action. Moreover, they appear to establish a functional linkage between the RodA dependent

elongasome machinery and outer membrane biosynthesis which is entirely novel. I am very supportive of this paper's eventual publication in Nature Communications and believe it could be a transformative paper. The authors present some very interesting and impactful data generally and the figures in particular, are excellent; quality.

However, I am critical of the manuscript with respect to its ability to relate to a wider scientific audience consistent with Nature communications rather than a specialist microbiology community which at present, this paper is clearly designed for. Some additional explanation is required in the text to ensure that the paper is accessible to a wide audience who may not be familiar with this research group's impressive and detailed track record in this field of microbiology. This is required in some detail to understand the context of what is written here. A good exemplar of this deficiency is the central experimental approach of growth in the presence of absence of ceftriaxone with the transposon library created. There is absolutely no explanation as to why that antibiotic is chosen or has relevance in this context!

I trust the authors will find my comments helpful in the development of their paper as they are intended to be constructive criticism

We have indicated the rationale for choosing ceftriaxone in the first paragraph of the "Results" section:

"The β -lactam ceftriaxone was chosen for our analysis since this broad spectrum 3rd generation cephalosporin is highly effective in inactivating all D,D-transpeptidases belonging to the PBP family and is specific of these enzymes as L,D-transpeptidases are not inactivated due to a combination of ineffective acylation and formation of an acyl-enzyme prone to hydrolysis²⁵⁻²⁶. For both the -CRO and +CRO conditions, IPTG and L-arabinose were added..."

Abstract

Line 14. "bypass" this is not a useful term in the context of the abstract. One presumes that authors mean "functional replacement" of PBPs by LD-transpeptidases to enable peptidoglycan biosynthesis. Whilst I appreciate the authors have used this term before in the literature some additional context/explanation is required to improve the accessibility of the paper. I also think that the three sentences from line 15 to 21 that summarise the major results of the paper are somewhat disjointed and do not naturally produce a narrative that endorses the final sentence of the abstract. In short, they authors need a better abstract and this a weakness of the paper as present.

We have rephrased the abstract as requested by the reviewer. "An unexpected impact" in the title has been replaced by "unexpected effects" to adhere to the terms used in the revised abstract:

Previous title:

Genome-wide identification of genes required for alternative peptidoglycan cross-linking in *Escherichia coli* revealed an unexpected impact of β -lactams

Revised title:

Genome-wide identification of genes required for alternative peptidoglycan cross-linking in *Escherichia coli* revealed unexpected effects of β -lactams

Previous abstract:

The D,D-transpeptidase activity of penicillin-binding proteins (PBPs) is the well-known primary target of β -lactam antibiotics that block peptidoglycan polymerization. β -lactam-induced killing involves complex downstream responses whose causes and consequences are difficult to deconvolute. Here we show that bypass of PBPs by a β -lactam-insensitive L,D-transpeptidase provides a powerful tool for studying these responses in living bacteria. Tn-seq analysis identified 179 genes selectively essential to prevent the toxic effects of PBP inactivation by β -lactams. Integrity of peptidoglycan synthesis mediated by RodA in a functional elongasome was essential as well as many genes involved in the synthesis of outer membrane polymers. Growth in the presence of β -lactams prevented the anchoring of the Braun lipoprotein to peptidoglycan and weakened the permeability barrier of the outer membrane. These findings reveal unsuspected side effects of β -lactams and identify a critical role of the outer membrane for homeostasis of the cell envelope in the presence of β -lactams.

Revised abstract:

The D,D-transpeptidase activity of penicillin-binding proteins (PBPs) is the well-known primary target of β -lactam antibiotics that block peptidoglycan polymerization. β -lactam-induced bacterial killing involves complex downstream responses whose causes and consequences are difficult to resolve. Here, we used the functional replacement of PBPs by a β -lactam-insensitive L,D-transpeptidase to identify genes essential to mitigate the effects of PBP inactivation by β -lactams in actively dividing bacteria. The functions of the 179 conditionally essential genes identified by this approach extended far beyond L,D-transpeptidase partners for peptidoglycan polymerization to include proteins involved in stress response and in the assembly of outer membrane polymers. The unsuspected effects of β -lactams included loss of the lipoprotein-mediated covalent bond that links the outer membrane to the peptidoglycan, destabilization of the cell envelope in spite of effective peptidoglycan cross-linking, and increased permeability of the outer membrane. The latter effect indicates that the mode of action of β -lactams involves self-promoted penetration through the outer membrane.

Line 34. I Suggest that the word subsequently is inserted after “chains that are“.

The text has been modified as suggested.

Line 36. The word “cleave is misleading and technically incorrect. The terminal D-ala of the pentapeptide is not removed as implied as some form of catabolic process but as a consequence of the mechanism of formation of a PEP-peptidoglycan intermediate that can be broken down in a number of ways.

The misleading statement:

“The latter enzymes, the D,D-transpeptidases, also referred to as penicillin-binding proteins (PBPs) as they are the essential targets of β -lactam antibiotics, cleave the D-Ala⁴-D-Ala⁵ peptide bond of pentapeptide stems (hence the D,D designation) and form 4→3 cross-links (Fig. 1c; Supplementary Fig. S1a)².”

has been modified as follows:

“The latter enzymes, the D,D-transpeptidases, are also referred to as penicillin-binding proteins (PBPs) as they are the essential targets of β -lactam antibiotics. For peptidoglycan polymerization, PBPs interact with the D-Ala⁴-D-Ala⁵ extremity of a pentapeptide stem (hence the D,D designation) and form a covalent link between D-Ala⁴ and their active-site Ser residue with the concomitant release of D-Ala⁵. In the following step, the resulting acyl-enzyme reacts with the second substrate, most often a tetrapeptide stem, to form a 4→3 cross-linked Tetra-Tetra dimer with the concomitant release of the PBP (Fig. 1c; Supplementary Fig. S1a)².”

Line 37. Do we need an arrow between 4-3?

The arrow is a common nomenclature used for this type of peptidoglycan bonds. We strongly believe that the arrow is important to indicate the direction of the amide bond.

Line 42. We have the word “Bypass” again but this time with a hyphen which is inconsistent. Whilst reference 8 is used to guide the reader, I would argue that their paper PLoS One 2013 Jul 4;8(7):e67831 actually provides a more useful and earlier description of “bypass”

We removed the hyphen in “bypass” and added the historical reference for this mechanism of β -lactam resistance.

Line 42 “unrelated” in what way? I don’t believe this term is helpful to the reader, comprehension of science and is potentially misleading.

We meant “structurally unrelated”. We think that it is important to mention that transpeptidases of the D,D and L,D specificities belong to evolutionary distinct protein families.

Line 58. “...for the rescue of PBPs by the 3-3...”. The subject is peptidoglycan biosynthesis which is rescued by the LD-transpeptidases. The function of PBPs might be rescued in this context but not the PBPs since they are rendered useless by the beta lactam drugs.

We added a clarification by replacing “... for the rescue of PBPs...” with “... for the rescue of the β -lactam-inactivated 4→3 cross-linking activity of PBPs...”

Line 62. “Evolvability” ie capacity of a system for adaptive evolution. “is this what the authors actually mean?”

The reviewer is right. This is exactly what we meant for “evolvability”, which is adapted to the plasticity and interplay of the assembly of peptidoglycan and other envelope polymers documented by the β -lactam resistance bypass mechanism under study.

Results

Line 85. The authors need to provide some more background with a simple sentence on the strain in question. Inspection of Table S1 reveal nothing about this strain and reference 23 appears to be a methods paper not related to the strain.

(i) We apologize for the mistake in the strain designation. “BW15113” was replaced by “BW25113” in the revised version of the main text and of Supplementary Table S1 that provides the accurate reference.

(ii) Reference 23 was mentioned for the Tn-seq and was accordingly moved at the appropriate position of the sentence.

(iii) We have added one sentence and one reference for introducing the relevant properties of the *E. coli* strain BW25113(*ycbB*, *relA*’). “...in the genome of *E. coli* BW25113(*ycbB*, *relA*’)⁸. This strain combines high level production of L,D-transpeptidase YcbB and of (p)ppGpp synthetase RelA’ upon induction by IPTG and L-arabinose of the *ycbB* and *relA*’ genes, respectively.

Is the strain used BW25513(*ycbB*, *relA*’) as in table S1? (Not BW15113 as stated..?)

We apologize for these mistakes (see answer above).

Line 86. Similarly the Authors should provide a clear rationale why ceftriaxone is used in this study and its relevance to PBP and LD-TP inhibitions as it is not apparent to a non-expert in beta-lactam development.

As detailed in the answer to the general comment (above), we have added a sentence Lines 94-98 describing the rationale for choosing ceftriaxone among β -lactams.

Line 86. Is a central tenet of the experiment that over expression of YcbB should overcome the inhibition of this enzyme by ceftriaxone produced from the chromosomal gene copy? If so, please make this clear.

In the 1st revised paragraph in the results section of the revised manuscript (see answer to the general comment of the reviewer), we have indicated that ceftriaxone is not an effective inhibitor of L,D-transpeptidases. Overproduction of YcbB from a plasmid copy of the *ycbB* gene is required because the chromosomal copy of the *ycbB* gene is not expressed at a sufficient level to sustain peptidoglycan synthesis. We would also like to underscore that overexpression of *ycbB* is also required for ampicillin resistance (ampicillin does not inhibit LDTs at concentration as high as 2,000 μ g/ml). Appropriate discussion of the underlining mechanism can be found in Triboulet et al. PLoS One 2013 (reference 25 of the revised manuscript).

Line 119-121. Many of the Fts genes listed are essential under any condition, so to try and

describe this in the context of + and - antibiotic is misleading. Whatever the authors are trying to say it needs rewording. The last sentence from line 122 is a profound finding however!

We respectfully disagree with this comment. The Tn-seq analyses also identified genes that are essential in both conditions and it is important to distinguish these genes from those that are selectively required in the +CRO condition.

lines 139-142. The authors state that ceftriaxone will inactivate PBP2 and cite their paper (Ref 8) and that of Erin Carlson (ref 32). The latter suggests that the primary affinity target of ceftriaxone by at least 20 fold is PBP3 in E.coli not PBP2. Thus, this statement needs to be held in context of the concentration of antibiotic used in the experiment. Can they say categorically that PBP2 is inactivated under these conditions? If so, what of YcbB? Ceftriaxone also inhibits this enzyme I believe and is a central tenet of their paper from 2013: PLoS One 2013 Jul 4;8(7):e67831.doi: 10.1371/journal.pone.0067831.

In reference 8 we showed that the peptidoglycan does not contain 4→3 cross-links indicating that all PBPs, including PBP2, are inactivated by β-lactams. In reference 35 of the revised manuscript (was ref 32), IC₅₀s were determined to be 0.3 μg/ml, 0.2 μg/ml, 0.2 μg/ml, and 0.01 μg/ml for the four D,D-transpeptidases of *E. coli* (PBP1a, PBP1b, PBP2, and PBP3, respectively). Thus, the reviewer is correct in stating that the concentration required for inhibition of PBP2 is 20-fold higher than that required for inhibition of PBP3 (0.2 μg/ml versus 0.01 μg/ml). However, all these concentrations are far below the concentration of 8 μg/ml used in our study. The central tenet of our PLoS One paper is that the acyl-enzymes resulting from inactivation of LDTs by penams (such as ampicillin) and cepheims (such as ceftriaxone) are prone to hydrolysis, in contrast to those resulting from inactivation of LDTs by carbapenems (such as meropenem or imipenem). This results in the absence of inhibition of peptidoglycan cross-linking by penams and cepheims and to resistance to these drugs at concentrations of 8 μg/ml or 16 μg/ml used in this study and in references 8 and 26.

These observations indicate that our conclusions are justified. We strongly believe that adding a comparison of the β-lactam concentrations required to inhibit LDTs and PBPs will unnecessarily increase the complexity of the manuscript.

Line 199. We need a reference to the dispensability of dapF at the end of the sentence that describes it.

We have added the reference 43 (was ref 40) at the indicated position as requested.

Line 201-210. The authors postulate that PBPs are able to cross link stem peptides form with LL-DAP instead of meso-(DL)-DAP and provide reference to examples in the literature that support LL-DAP dependent growth. Patin et al <https://pubmed.ncbi.nlm.nih.gov/20659527/> report that the Km of Meso-DAP for MurE is 40mM. Thus the 3000-fold difference in concentration to allow LL-DAP incorporation postulated in the manuscript here is 120mM. This seems unlikely is in 10 fold higher than that reported by Mengin-Lecreulx et la J

Bacteriol 1988 May;170(5):2031-9 who also discuss “The low extent of cross linkage of the peptidoglycan isolated from the dapF mutant suggested that the presence of an LL-DAP residue interfered in some way or another with this late step of peptidoglycan biosynthesis”. Moreover whilst LL-DAP incorporation into E.coli peptidoglycan is possible in a laboratory setting it is clear that the preferred substrate for peptidoglycan formation is predominantly Meso-DAP. Cathewood et al 2020 (<https://pubmed.ncbi.nlm.nih.gov/32048840/>) provides an alternative explanation in that in the presence of LL-DAP as a donor in the transpeptidase reaction E.coli PBP1b switches to formation of DD-carboxypeptidase activity predominantly, i.e. the formation of tetrapeptides which are the required substrates for LD-transpeptidases including YcbB. If this stereochemical bias is also found in other class A and class B PBPs then this may account for the observations seen. It would be interesting to know if the crosslinking composition of PG from cells lacking DapF.

We thank the reviewer for these interesting remarks and acknowledge the fact that the first version of the section on DapF was inaccurate with respect to previous work. We have rewritten the entire paragraph and cited the references.

Submitted manuscript:

Consequences of replacement of *meso*DAP by L,L-DAP in PG precursors on resistance and envelope synthesis. DapF, which converts L,L-DAP into *meso*DAP, was reported to be dispensable for growth in *E. coli*. MurE ligase catalyzes the addition of L,L-DAP to UDP-MurNAc-L-Ala-D-Glu, albeit less effectively than the addition of *meso*DAP due to a 3,000-fold difference in K_m ⁴⁰. Since epimerization of L,L-DAP by DapF is the only source of *meso*DAP in *E. coli*, this implies that PBPs are able to cross-link stem peptides containing L,L-DAP instead of *meso*DAP. Concordantly, *dapF* was unessential in the -CRO condition. On the contrary, no transposon insertions were found in this gene in the +CRO condition. Lethality was not due to the inability of YcbB to cross-link L,L-DAP-containing stem peptides since 3→3 cross-linked dimers were detected in a Δ *dapF* mutant (Supplementary Fig. S4). Likewise, deletion of *dapF* did not prevent the anchoring of the Braun lipoprotein to the PG. Thus, L,D-transpeptidases, similarly to PBPs, tolerated the replacement of *meso*DAP by L,L-DAP. The essentiality of *dapF* in the +CRO condition may therefore be accounted for by a reduced efficacy of the PG assembly pathway, as observed above for the *ldcA* null mutant.

Revised manuscript:

Consequences of replacement of *meso*DAP by L,L-DAP in PG precursors on resistance and envelope synthesis. DapF, which converts L,L-DAP into *meso*DAP, was reported to be dispensable for growth in *E. coli*. MurE ligase catalyzes the addition of L,L-DAP to UDP-MurNAc-L-Ala-D-Glu, albeit less effectively than the addition of *meso*DAP due to a 3,000-fold difference in K_m ⁴³. Deletion of the gene encoding *dapF* results in the incorporation of L,L-DAP into peptidoglycan precursors. L,L-DAP-containing stem peptides were ineffectively used as acyl donors by the PBPs but not as acceptor⁴³. The overall cross-linking of peptidoglycan was moderately reduced⁴³. Concordantly, L,L-DAP was strongly discriminated against *meso*DAP by purified PBP1b *in vitro*⁴⁴. In our study, *dapF* was non-essential in the -CRO condition. On the contrary, no transposon insertions were found in this gene in the +CRO condition

(Supplementary Data file 1b; Supplementary Data file 2). Formation of 3→3 cross-linked dimers was detected in a $\Delta dapF$ mutant (Supplementary Fig. S4) but mass spectrometry analysis did not enable discriminating between the two DAP stereoisomers. Our peptidoglycan analysis additionally showed that deletion of *dapF* did not prevent the anchoring of the Braun lipoprotein to the PG (Supplementary Fig. S4). Thus, L,D-transpeptidases, similarly to PBPs, tolerated to a certain extent the incorporation of L,L-DAP into peptidoglycan precursors following deletion of *dapF*. The essentiality of *dapF* in the +CRO condition might be accounted for by a reduced efficacy of the PG assembly pathway, as observed above for the *ldcA* null mutant.

Line 216-221. If “all but one (of the) colonic acid biosynthetic enzymes, Wca were (are) specific essential in the +CRO conditions”, how is the next sentence correct? This entire section from line 216 is difficult to follow and understand the logic of.

The beginning of this section has been rephrased to improve clarity.

Submitted version:

Accumulation of capsule intermediates abolished YcbB-mediated β -lactam resistance.

The outermost polymer of the cell wall of most *E. coli* strains is a capsule made of colanic acid⁴¹. All but one colanic acid biosynthetic enzymes, WcaJ, were specifically essential in the +CRO condition (Fig. 4b, 4c, and 4d). WcaJ catalyzes the first committed step in the assembly of the colanic acid precursor, indicating that this polymer is unessential for growth in the presence of ceftriaxone. Given that the loss of all other colanic acid biosynthesis enzymes is expected to result in accumulation of undecaprenyl-linked precursors, we concluded that these enzymes are essential for preventing the sequestration of the lipid transporter. According to this model, sequestration of C₅₅ would indirectly inhibit the assembly of other polymers, such as the essential PG, that rely on the same lipid carrier. As confirmation, we showed that deletion of the gene encoding WcaJ enabled growth of a WcaI mutant in the presence of ceftriaxone (Supplementary Fig. S6a). Combined, these results indicate that the capsule is, in itself, unessential for growth in the presence of ceftriaxone although biosynthetic enzymes are essential to prevent reduced PG synthesis by sequestration of the lipid carrier.

Revised version:

Accumulation of capsule intermediates abolished YcbB-mediated β -lactam resistance.

The outermost polymer of the cell wall of most *E. coli* strains is a capsule made of colanic acid⁴⁵. WcaJ catalyzes the first committed step in the assembly of the colanic acid precursor. The gene encoding WcaJ was non-essential in the -CRO and +CRO conditions, indicating that the absence of the capsule was compatible for growth both in the absence or presence of ceftriaxone (Fig. 4b, 4c, and 4d). Unexpectedly, all other enzymes in the colanic acid synthesis pathway were essential in the +CRO but not in the -CRO conditions. Given that the loss of all other colanic acid biosynthesis enzymes is expected to result in accumulation of undecaprenyl-linked precursors, we concluded that these enzymes are essential in the +CRO condition for preventing the sequestration of the lipid carrier. According to this model, sequestration of C₅₅ would indirectly inhibit the assembly of other polymers, such as the essential PG, that rely on the same lipid carrier. As confirmation, we showed that deletion of the gene encoding WcaJ,

catalyzing the first committed step of the colanic acid precursor synthesis, restored growth of a WcaI mutant in the presence of ceftriaxone (Supplementary Fig. S6a). Combined, these results indicate that the capsule is, in itself, non-essential for growth in the presence of ceftriaxone although biosynthetic enzymes are essential to prevent reduced PG synthesis by sequestration of the lipid carrier.

Line 242. Again, this section appears hard to reconcile and depends how the experiment has been conducted. The lack of essentiality in individual mutations by TRADIS in the genes responsible for ECA polymer formation suggests functional redundancy between the three ECA polymer systems surely. The result that many of these genes become essential under the +CRO condition indicates a connection between LD transpeptidases and the PBPs with outer membrane integrity.

For ECAs, the first gene involved in the assembly of the common precursor (*wecA*) is essential only in the +CRO condition. This essentiality indicates that at least one of the three ECA polymers (ECA_{PGL}, ECA_{LPS}, and ECA_{CYC}) is required. ECA_{LPS} and ECA_{CYC} are dispensable for ceftriaxone resistance since deletion of genes specifically involved in synthesis of one of these two polymers was possible (*waaL* or *wzzE*; Fig. 5c). We also constructed a double mutant ($\Delta waaL \Delta wzzE$; producing only the ECA_{PGL}), which was resistant to ceftriaxone, indicating the absence of redundancy between the ECA_{LPS} and ECA_{CYC} for the resistance phenotype (Supplementary Fig. S6b). A short sentence has been added in this section to underline this lack of redundancy: "...growth in the presence of ceftriaxone (Fig. 5c; Supplementary Fig. S6b). The phenotype of the double mutant indicates the absence of redundancy between the ECA_{LPS} and ECA_{CYC} for ceftriaxone resistance. By elimination, ...". These results indicate that the ECA_{PGL} is essential in the +CRO condition although this could not be directly demonstrated since the gene encoding the ECA_{PGL} synthase has not been identified. As indicated in the answer to comment 5 from reviewer #1, the gene encoding the ECA_{PGL} synthase is likely to be one of the 14 genes of unknown function selectively essential in the +CRO condition.

Line 272-282. The logic in this section is difficult to follow and a prerequisite appears to be an encyclopaedic knowledge of peptidoglycan chemistry and its relationship to LD transpeptidase required for Brauns lipoprotein anchoring (LdtA-C) as opposed to 3-3 crosslink formation (LdtD, LdtE). The fact that LdtD and YcbB are the same is not helpful. None of this is explained to enable to reader to understand the argument being made and needs to be rewritten to provide that basis.

This section has been rephrased to improve clarity as requested.

Submitted version

Growth with β -lactams impacts the anchoring of the Braun lipoprotein to peptidoglycan.

The Tn-seq analysis showed that the Braun lipoprotein is dispensable for growth in both -CRO and +CRO conditions. Analyses of the PG structure (Supplementary Fig. S7) revealed that growth in the presence of ceftriaxone dramatically reduced the anchoring of the Braun lipoprotein to PG. Thus, ceftriaxone may inhibit the anchoring reaction catalyzed by LDTs

(Supplementary Fig. S1c). Competition of LDTs for the same mucopeptide donor, as previously proposed²², may also contribute to limiting the anchoring of the Braun lipoprotein to PG. However, this appears less likely since the high proportion of 3→3 cross-links in the -CRO conditions (40%) was not detrimental to the anchoring of the lipoprotein (Supplementary Fig. S7).

Revised version

Growth with β -lactams impacts the anchoring of the Braun lipoprotein to peptidoglycan.

The *E. coli* genome encodes six members of the LDT protein family (Supplementary Fig. S1). YcbB and YnhG catalyze formation of 3→3 cross-links, whereas ErfK, YbiS, and YcfS are responsible for covalently linking the Braun lipoprotein (Lpp) to PG^{10,11}. The remaining enzyme, YafK, hydrolyzes the Lpp-PG link formed by ErfK, YbiS, and YcfS, thereby releasing the lipoprotein^{22,23}. The Tn-seq analysis showed that the Braun lipoprotein is dispensable for growth in both -CRO and +CRO conditions. Analyses of the PG structure revealed that growth in the presence of ceftriaxone dramatically reduced the anchoring of the Braun lipoprotein to PG (Supplementary Fig. S7). Western blot analyses showed that Lpp synthesis was not reduced in the +CRO condition (Supplementary Fig. S7g). Thus, ceftriaxone may inhibit the anchoring reaction catalyzed by LDTs (Supplementary Fig. S1), but this inhibition is unlikely to be direct since LDTs are not effectively inhibited by β -lactams, except those belonging to the carbapenem class^{25,26}.

Any hypothesis on the mode of inhibition of Lpp anchoring to PG by β -lactams should take into consideration the fact that LDTs for PG cross-linking (YcbB and YnhG) and for Lpp anchoring (ErfK, YbiS, and YcfS) interact with the same donor substrate, a tetrapeptide donor stem, in the first catalytic step common to the two types of L,D-transpeptidation reactions (Supplementary Fig. S1b and S1c)¹². In the second step, the resulting acyl enzymes react either with a peptidoglycan stem to form 3→3 cross-links or with Lpp to covalently link the lipoprotein to PG. The common substrate of the first step of the transpeptidation reactions implies that competition of LDTs for the same tetrapeptide donor, as previously proposed²², may contribute to limiting the anchoring of the Braun lipoprotein to PG. However, this appears unlikely in the context of YcbB-mediated ceftriaxone resistance since the high proportion of 3→3 cross-links in the -CRO conditions (40%) was not detrimental to the anchoring of the lipoprotein (Supplementary Fig. S7). Thus, impaired Lpp anchoring in the presence of ceftriaxone may result from another indirect inhibitory effect of ceftriaxone on the activity of ErfK, YbiS, and YcfS, such as the formation of their tetrapeptide substrate by a partner enzyme, or the stimulation of the release of the lipoprotein by YafK.

Lines 283-300. Arguably the best written section of this latter part of the manuscript. It would have been perfect with just one additional sentence that made clear that the CPRG assay assesses outer membrane integrity by detection of LacZ enzyme activity in the media as a result of leakage (or words to that effect).

We have indicated in this section that there is no specific transporter for CPRG. Thus, hydrolysis of this compound is only possible if the permeability barrier of the envelope is compromised:

“Thus, exposure to subinhibitory concentrations close to the minimal inhibitory concentration (MIC) of the drug resulted in increased permeability as there is no specific transporter for CPRG”

Lines 318-329. Whilst the concluding sentence of this paragraph appears to be supported by the paper, the preceding lines require an expert level appreciation of the most recent literature in order to interpret. Also do the authors actually mean what is said regarding pre-synthesis of 3-3 crosslinked strands which are then inserted into preexisting PG. Ho is this mediated, further 3-3 crosslinking? This will not do.

We agree that this section introduces complex notions on the mode of peptidoglycan polymerization that are supported by very recent work. Meanwhile, we think that it is important to underscore that accumulation of toxic peptidoglycan polymers in response to PBP2 inactivation is experimentally documented. It is also important to point out that YcbB is able to detoxify these polymers through a unique mode of peptidoglycan polymerization, which is also experimentally documented by pulse-chase experiments. Therefore, we would like to maintain this section without providing a detailed description of the polymerization model and of the underlying experimental approaches that can be found in the cited references. We have carefully reconsidered this section and have introduced the following minor modifications for improving clarity:

“In susceptible bacteria, accumulation of uncross-linked glycan strands following PBP2 inactivation by β -lactams is deleterious and their degradation by the lytic transglycosylase SltY contributes to cell survival²⁰. In contrast, inactivation of SltY favors YcbB-mediated resistance to β -lactams, with uncross-linked glycan strands formed by the elongasome complex serving as suitable substrates for the assembly of a functional PG polymer⁷.”

REVIEWERS' COMMENTS

Reviewer #1 (Remarks to the Author):

The authors have addressed my comments in great detail and I have no further concerns.

Reviewer #2 (Remarks to the Author):

The authors have significantly improved the manuscript in response to the previous reviews and have satisfactorily addressed my concerns. Very nice, significant, and well-done study.